# RLHF Workflow: From Reward Modeling to Online RLHF

**A Comprehensive Practical Alignment Recipe of Iterative Preference Learning**

**Hanze Dong**[1*]     **Wei Xiong**[2*]     **Bo Pang**[1*]     **Haoxiang Wang**[2*]

**Han Zhao**[2]     **Yingbo Zhou**[1]     **Nan Jiang**[2]     **Doyen Sahoo**[1]

**Caiming Xiong**[1†]     **Tong Zhang**[2†]

[1]*Salesforce AI Research*     [2]*University of Illinois Urbana-Champaign*

**Reviewed on OpenReview:** `https://openreview.net/forum?id=a13aYUU9eU`

## Abstract

We present the workflow of Online Iterative Reinforcement Learning from Human Feedback (RLHF) in this technical report, which is widely reported to outperform its offline counterpart by a large margin in the recent large language model (LLM) literature. However, existing open-source RLHF projects are still largely confined to the offline learning setting. In this technical report, we aim to fill in this gap and provide a detailed recipe that is easy to reproduce for online iterative RLHF. In particular, since online human feedback is usually infeasible for open-source communities with limited resources, we start by constructing preference models using a diverse set of open-source datasets and use the constructed proxy preference model to approximate human feedback. Then, we discuss the theoretical insights and algorithmic principles behind online iterative RLHF, followed by a detailed practical implementation. Our trained LLM achieves impressive performance on LLM chatbot benchmarks, including AlpacaEval-2, Arena-Hard, and MT-Bench, as well as other academic benchmarks such as HumanEval and TruthfulQA. We have shown that supervised fine-tuning (SFT) and iterative RLHF can obtain state-of-the-art performance with fully open-source datasets. Further, we have made our models, curated datasets, and comprehensive step-by-step code guidebooks publicly available.

## 1    Introduction

*Reinforcement Learning from Human Feedback* (RLHF) (Christiano et al., 2017; Ziegler et al., 2019) has become as a key technique for integrating human preference signals into machine learning methods. In particular, RLHF has become a standard component in the post-training pipe line of foundation Large Language Models (LLMs), which serves to align the outputs of these models with human values such as helpfulness, harmlessness, and honesty (Ouyang et al., 2022; Bai et al., 2022a). Notable examples include the revolutionary closed-source ChatGPT (OpenAI, 2023), Claude (Anthropic, 2023), and Gemini (Team et al., 2023), as well as the powerful open-source models like Zephyr (Tunstall et al., 2023), Starling (Zhu et al., 2023), and LLaMA-3 (Meta, 2024). In particular, since the introduction of ChatGPT, RLHF has attracted significant interest in a diverse set of communities. However, compared to the supervised fine-tuning that is rather well studied with many great open-source projects like Open-Hermes (Teknium, 2023) and Vicuna (Zheng et al., 2023), RLHF remains relatively under-explored within the open-source community.

---

*The first four authors are core contributors to this project listed in random order. The full authorship contribution statements are provided in Appendix A. Email: {`hanze.dong, b.pang, yingbo.zhou, dsahoo, cxiong`}`@salesforce.com`, {`wx13,`
`hwang264, hanzhao, nanjiang, tozhang`}`@illinois.edu`.

†Project leads.

To facilitate our discussion, we build upon the standard RLHF workflow (Ouyang et al., 2022; Bai et al., 2022b; Touvron et al., 2023). We characterize an LLM by a policy $\pi$, which takes a prompt $x \in \mathcal{X}$ and produces a response $a \in \mathcal{A}$ from the distribution $\pi(\cdot|x)$. We denote the initial model of RLHF as $\pi_0$, which is fine-tuned on some instruction-following data after the pre-training stage. We assume that we have a prompt set that is sampled from some unknown but fixed distribution $x \sim d_0$.

The central principle of RLHF is to learn from *relative feedback*, rather than an absolute reward signal, as is common in traditional RL literature. This approach is preferred because human raters often struggle to provide accurate absolute ratings; instead, they find it easier to choose between two options, indicating which response they prefer (Christiano et al., 2017). This idea can further date back to the study of dueling bandit (Joachims et al., 2007; Yue et al., 2012) in the context of online decision making. Specifically, we assume that we have access to a *Preference Oracle*, that is a proxy of a real-world human rater. Mathematically, we have the following formal definition.

**Definition 1** (Preference Oracle)**.** *There exists a preference oracle* $\mathbb{P} : \mathcal{X} \times \mathcal{A} \times \mathcal{A} \rightarrow [0, 1]$*, and we can query it to receive the preference signal:*

$$y \sim \text{Ber}\big(\mathbb{P}(a^1 \succ a^2 | x, a^1, a^2)\big),$$

*where* $\text{Ber}(t)$ *is a Bernoulli distribution with parameter* $t$ *and* $y = 1$ *means* $a^1$ *is preferred to* $a^2$*, and* $y = 0$ *means that* $a^2$ *is preferred.*

To further simplify the problem, it is commonly assumed that the preference signal can be modeled using the reward-based Bradley-Terry model, a well-known approach in preference learning (Bradley & Terry, 1952; Ouyang et al., 2022; Bai et al., 2022a; Touvron et al., 2023).

**Definition 2** (Bradley-Terry Model)**.** *There exists a ground-truth reward function* $r^*$ *and the preference model satisfies:*

$$\mathbb{P}(a^1 \succ a^2 | x, a^1, a^2) = \frac{\exp(r^*(x, a^1))}{\exp(r^*(x, a^1)) + \exp(r^*(x, a^2))} = \sigma\big(r^*(x, a^1) - r^*(x, a^2)\big), \tag{1}$$

*where* $\sigma(z) = 1/(1 + \exp(-z))$ *is the sigmoid function.*

Although the BT model may not fully capture the complex human preference, it tends out to be a useful approximation to connect the learning objective of RLHF with reward maximization and has achieved tremendous success in making ChatGPT (Ouyang et al., 2022) and Claude (Bai et al., 2022a). Meanwhile, in response to the imperfect nature of BT model, the goal in this RLHF formulation is to optimize the following KL-regularized target:

$$J(\pi) = \mathbb{E}_{x \sim d_0} \mathbb{E}_{a \sim \pi(\cdot|x)} \left[ r^*(x, a) + \eta \log \frac{\pi_0(a|x)}{\pi(a|x)} \right] = \mathbb{E}_{x \sim d_0} \left[ \mathbb{E}_{a \sim \pi(\cdot|x)}[r^*(x, a)] - \eta D_{\text{KL}}(\pi(\cdot|x) \| \pi_0(\cdot|x)) \right], \tag{2}$$

where $\eta > 0$ is the KL penalty coefficient. This formulation is widely studied in practice (Ziegler et al., 2019; Wu et al., 2021; Ouyang et al., 2022; Rafailov et al., 2023; Liu et al., 2023a; Xiong et al., 2023) and admits the following intractable closed-form solution (Zhang, 2023)

$$\pi^*(a|x) = \frac{1}{Z(x)} \pi_0(a|x) \exp\left(\frac{1}{\eta} r^*(x, a)\right), \tag{3}$$

where $Z(x) = \sum_{a' \in \mathcal{A}} \pi_0(a'|x) \exp\left(\frac{1}{\eta} r^*(x, a')\right)$ is the normalization constant.

In the subsequent subsections, we first describe the existing approaches, and discuss their challenges, which should serve as the motivation for our project.

## 1.1 Previous RLHF Algorithms and Their Challenges

Generally, previous RLHF methods can be largely divided into two categories: (1) deep RL-based approach using Proximal Policy Optimization (PPO) (Schulman et al., 2017; Christiano et al., 2017; Ziegler et al.,

2019) and (2) (offline) direct preference learning (e.g., DPO) approaches (Zhao et al., 2023; Rafailov et al., 2023; Azar et al., 2023; Tang et al., 2024).

**DRL-based framework.** The DRL-based framework consists of two stages. In the first stage, a reward model is trained. Specifically, given a preference dataset $\mathcal{D}_{\text{off}} = \{(x, a^w, a^l)\}$, where $a^w$ is a response preferred over $a^l$ given the instruction or prompt $x$. The log-likelihood function of the BT model can be expressed as:

$$\ell_{\mathcal{D}_{\text{off}}}(\theta) = \sum_{(x, a^w, a^l, y) \in \mathcal{D}_{\text{off}}} \log \left( \sigma \big( r_\theta(x, a^w) - r_\theta(x, a^l) \big) \right). \tag{4}$$

We can compute the maximum likelihood estimator (MLE) $r_{\text{MLE}}$ based on $\mathcal{D}_{\text{off}}$ by maximizing the $\ell_{\mathcal{D}_{\text{off}}}(\theta)$. In the second stage, DRL methods like PPO can be applied to optimize against the following reward:

$$\hat{r}(x, a) = r_{\text{MLE}}(x, a) - \eta \log \frac{\pi(a|x)}{\pi_0(a|x)}.$$

This approach has been employed by ChatGPT (Ouyang et al., 2022) and Claude (Bai et al., 2022a) and has contributed to the alignment of LLaMA-2 (Touvron et al., 2023). However, it is known that even in the best case, tuning the DRL method to its best performance requires extensive efforts in hyper-parameter selection and code-level optimization (Choshen et al., 2019; Engstrom et al., 2020). This becomes even more challenging in the context of LLMs, as fine-tuning LLMs is computationally expensive and searching the complicated hyper-parameters configuration is generally infeasible. Additionally, the PPO algorithm requires loading multiple LLMs simultaneously, including the actor (policy), critic (value network), reward model, and reference model (for KL estimation), which places significant pressure on GPU memory, especially for resource-constrained open-source projects.

**Direct preference learning.** In view of the above issues of PPO, there is an innovative line of work that directly learns from human preference datasets without explicitly constructing a reward function (Zhao et al., 2023; Rafailov et al., 2023; Azar et al., 2023). Among these methods, the direct preference optimization (DPO) algorithm is particularly popular. It leverages Equation 3 to formulate reward as a function of policy and directly optimizes the following loss function using the preference dataset $\mathcal{D}_{\text{off}}$:

$$\mathcal{L}_{\mathcal{D}_{\text{off}}}(\theta, \pi_0) = - \sum_{(x, a^w, a^l) \in \mathcal{D}_{\text{off}}} \left[ \log \sigma \Big( \eta \log \frac{\pi_\theta(a^w|x)}{\pi_0(a^w|x)} - \eta \log \frac{\pi_\theta(a^l|x)}{\pi_0(a^l|x)} \Big) \right]. \tag{5}$$

In the ideal case where there is no optimization error, the minimizer of Equation (5) is the same as the two-staged DRL framework (Rafailov et al., 2023; Azar et al., 2023). Meanwhile, direct preference learning algorithms are generally easier to tune and require fewer computational resources compared to DRL methods. Considering these factors, in our project, we focus on direct preference learning algorithms while leaving the study of the DRL-based framework for future research.

The open-source project Zephyr (Tunstall et al., 2023) serves as a milestone to popularize the DPO algorithm, where the authors provide a comprehensive guide for training LLMs through vanilla DPO and distillation from the teacher model ChatGPT. Following the Zephyr framework, many open-source models have been fine-tuned using vanilla DPO and their qualities are largely improved compared to their SFT counterparts. Impressively, on various LLM benchmarks' leaderboards, such as those reported by (Dubois et al., 2023; Zheng et al., 2023), models fine-tuned using DPO predominantly exhibit superior alignment and effectiveness.

While the vanilla offline direct preference learning algorithms are useful in some case studies, they also face certain challenges. Specifically, they are considered *offline* because they learn from an offline preference dataset collected prior to the alignment process. We can formulate this data collection process as:

$$x \sim d_0, a^1 \sim \pi_D^1, a^2 \sim \pi_D^2, \qquad y \sim \text{Ber}\big( \mathbb{P}(a^1 \succ a^2 | x, a^1, a^2) \big). \tag{6}$$

Here, $(\pi_D^1, \pi_D^2)$ represent two behavior policies, often taken as $\pi_0$, other open-sourced models or proprietary models. The term "offline learning" implies that we cannot further query the preference oracle $\mathbb{P}$ during the training process. However, the finite dataset $\mathcal{D}_{\text{off}}$ fails to cover the entire prompt-response space and the resulting policy model often performs poorly when faced with out-of-distribution data (Burns et al., 2023).

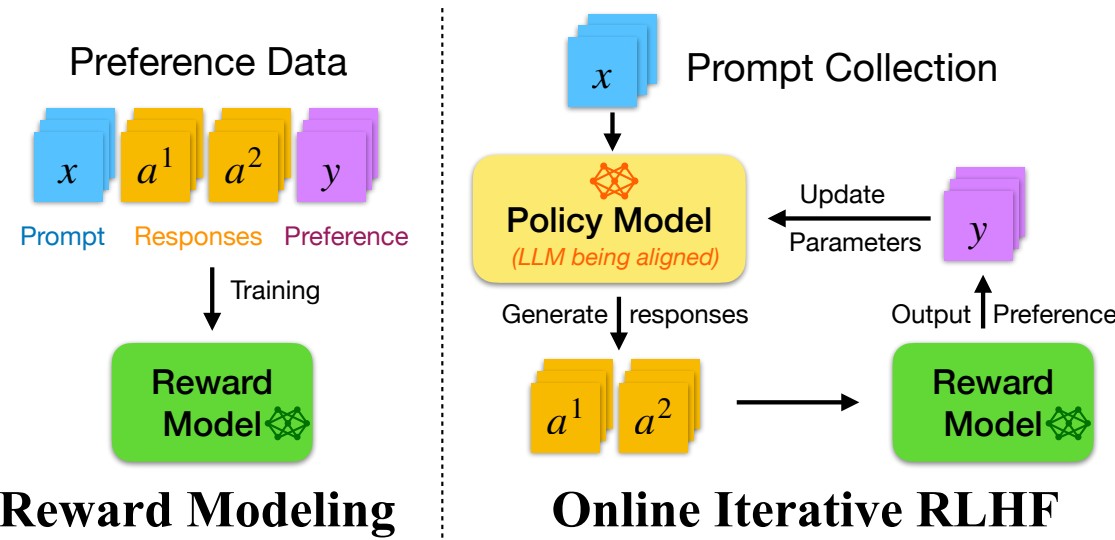

Figure 1: A simplified illustration of reward modeling and online iterative RLHF.

In particular, along the way of the RLHF training, the average density ratio $\frac{\pi(a|x)}{\pi_0(a|x)} > \exp(25)$ as reported in Figure 13 of Bai et al. (2022a). Therefore, the distribution shift between policies is usually very large, and it is unlikely that we can learn the optimal policy solely from a pre-collected dataset.

### 1.2 Online Iterative RLHF

In contrast, the Claude project (Bai et al., 2022a) and LLaMA-2 project (Touvron et al., 2023) have demonstrated that online iterative RLHF can significantly improve model performance. The process of online iterative RLHF, as formally formulated in Xiong et al. (2023), can be summarized as follows. Given the pre-collected preference dataset $\mathcal{D} = \mathcal{D}_{\text{off}}$ (if applicable, otherwise empty), for each iteration $t \in [T]$:

- we first update the policy pair $(\pi_t^1, \pi_t^2)$ based on the historical data $\mathcal{D}$ collected so far;

- we collect $m$ tuples as $\mathcal{D}_t$: sample a random prompt by $x_{t,i} \sim d_0$, collect two responses by $(a_{t,i}^1, a_{t,i}^2) \sim (\pi_t^1, \pi_t^2)$, and query the preference signal $y_{t,i} \sim \mathbb{P}$;

- update $\mathcal{D} \leftarrow \mathcal{D} \cup \mathcal{D}_t$.

The effectiveness of online data can be intuitively understood as continuously collecting new online data to strategically explore the prompt-response space and mitigating the out-of-distribution (OOD) issue. We also refer readers to Xiong et al. (2023) for a detailed theoretical explanation for online iterative RLHF. The hybrid formulation presented here is mainly for generality and is motivated by the LLaMA-2 project (Touvron et al., 2023). This formulation is also loosely related to some of the previous RL literature like Xie et al. (2021); Song et al. (2022).

We notice that the results presented in LLaMA-2 and Claude are based on the deep RL method, PPO, and the data, models, and training details are not fully accessible to the open-source community. Moreover, compared to the vanilla offline DPO, its online iterative version is still largely under-explored in the literature. Xiong et al. (2023) made the first step towards understanding the advantage of online exploration in DPO training from a theoretical perspective, and the main purpose of this work is to provide a detailed guidance to make the online iterative RLHF pipeline more accessible to the open-source community so that others can easily reproduce.

### 1.3 Human Feedback Approximation

Ideally, the online preference signal is sampled from a representative group of human labelers. However, human feedback is extremely expensive in practice, which the open-source community usually cannot afford. In the literature, there is a line of work showing that training a proxy preference model, and using the preference model to give proxy labels in a semi-supervised manner improve the model performance (Dong et al., 2023; Yuan et al., 2023; Liu et al., 2023a; Hoang Tran, 2024). We conjecture that this is because the reward model (discriminator) usually generalizes better than the policy (generator).

In particular, Hoang Tran (2024) shows that if the preference model (reward model) is trained on a diverse set of preference datasets, the Pair-RM (Jiang et al., 2023) with only 0.4B parameters can provide iterative preference learning with meaningful signals so that the resulting model[1] achieves an impressive AlpacaEval-2 length-control win rate of 26.4%. Motivated by this line of work, we first train a proxy preference (reward) model based on the diverse open-source preference datasets in Section 2 and then use the resulting model to provide preference signals for the subsequent iterative RLHF.

### 1.4 Related Work

We have presented many related works in the area of RLHF in the previous subsections. For completeness, we also include a more comprehensive literature review in this subsection.

**RLHF and RLHF algorithms.** The idea of learning from relative feedback could date back to the study of dueling bandit (Joachims et al., 2007; Yue et al., 2012) and the current RLHF framework was first popularized by Christiano et al. (2017), which served to direct the attention of the deep RL community to the preference-based feedback. Then, these techniques were further introduced to fine-tune LLMs for the summarization task (Ziegler et al., 2019; Stiennon et al., 2020). The dominant RLHF framework used for the modern LLM alignment was first thoroughly developed in Instruct-GPT (ChatGPT) (Ouyang et al., 2022), Claude (Bai et al., 2022a), and LLaMA-2 (Touvron et al., 2023) projects. These works typically involve constructing a reward model based on the MLE of the Bradley-Terry model, and then using the PPO algorithm to optimize the reward signals with KL regularization. One notable exception is that the LLaMA-2 uses a mixture of rejection sampling fine-tuning (Dong et al., 2023; Wang et al., 2024) and PPO in their RLHF pipeline. We refer the interested readers to Bai et al. (2022a); Touvron et al. (2023) for a detailed description. However, the use of PPO in RLHF has limitations. It is known to be unstable (Choshen et al., 2019), sensitive to implementation (Engstrom et al., 2020), and resource-intensive (Yuan et al., 2023). Despite some efforts to improve PPO in the context of RLHF (Li et al., 2023; Chan et al., 2024; Chang et al., 2024; Zhong et al., 2024), reproducing the successful results achieved with PPO is challenging for the open-source community due to these limitations as it requires extensive efforts and resources that the open-source communities usually cannot afford. In recognition of these issues of PPO, a line of work studies the (offline) direct preference learning algorithms, including Slic (Zhao et al., 2023), DPO (Rafailov et al., 2023), IPO (Azar et al., 2023), KTO (Ethayarajh et al., 2024), ARM (Pang et al., 2024), and GPO (Tang et al., 2024). These algorithms skip the reward modeling step, and optimize a designed loss target on the offline preference dataset directly (hence the name). It is widely observed that the direct preference learning algorithms are much more stable than the PPO, and achieve impressive performance evaluated by standard benchmarks (Tunstall et al., 2023; Dubois et al., 2023; Zheng et al., 2023).

Despite the advances offered by these direct preference learning algorithms, their implementation is typically offline and off-policy. This means they operate on preference datasets that were previously collected by other models—often powerful, proprietary teacher models like ChatGPT—before the training process begins (Tunstall et al., 2023; Cui et al., 2023).

**RLHF benefits from online (iterative) learning.** Roughly speaking, online iterative learning means that we will deploy the intermediate models and query human feedback for the responses of these models. Intuitively, this strategy can help to mitigate the OOD issue of the learned reward model (Gao et al., 2023), and its advantages have been reported in Ouyang et al. (2022); Touvron et al. (2023) for the PPO-based framework. Even when the additional feedback is derived from a proxy reward constructed from the same

---

[1] `https://huggingface.co/snorkelai/Snorkel-Mistral-PairRM-DPO`

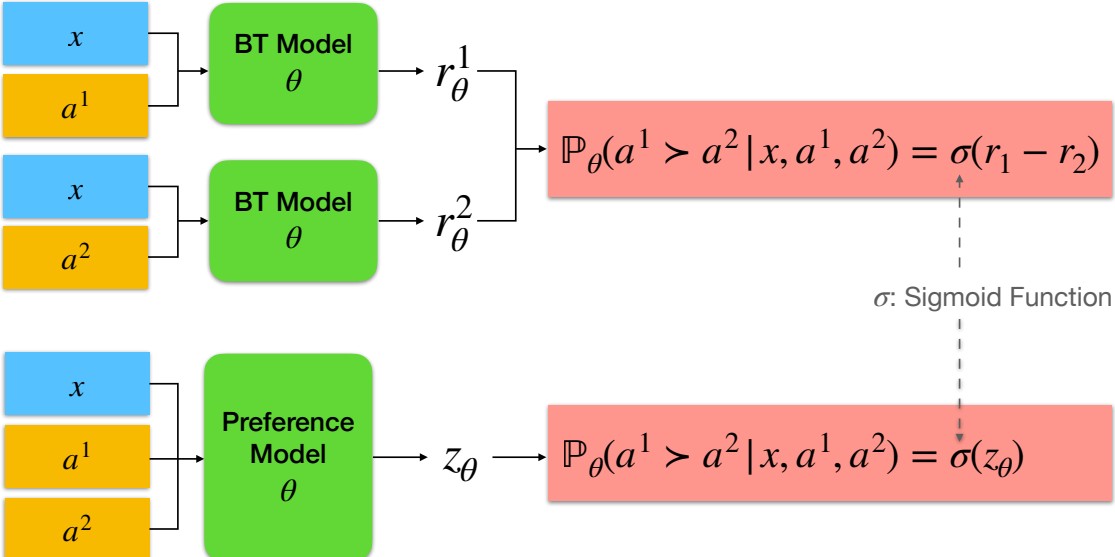

Figure 2: Illustration of the Bradley-Terry (BT) model and preference model.

offline dataset (similar to semi-supervised learning), iterative rejection sampling fine-tuning (RAFT) (Dong et al., 2023) and DPO based on samples from the target distribution estimator have been shown to outperform the original offline counterparts (Pang et al., 2024; Liu et al., 2023a). Furthermore, recent works (Xiong et al., 2023; Xu et al., 2023b; Hoang Tran, 2024; Yuan et al., 2024b; Swamy et al., 2024; Chen et al., 2024b; Ye et al., 2024; Guo et al., 2024; Rosset et al., 2024; Tajwar et al., 2024; Calandriello et al., 2024; Wu et al., 2024) have demonstrated that online iterative variants of direct preference learning algorithms significantly outperform their offline counterparts. In particular, we refer the interested readers to Guo et al. (2024) for the extensive experimental results with different offline base algorithms.

## 2 Reward Modeling as Human Feedback Approximation

We present the details of preference model construction in this section, where we study both the reward modeling as MLE of the BT model and the general preference model. We consider two versions of the training set: MIX1: HH-RLHF + SHP + UltraFeedback + Summarization, which is the mixture of dataset used to train the current state-of-the-art open-source model. (Stiennon et al., 2020), and MIX2: all the open-source datasets we collect, with details provided in Table 5.

### 2.1 Bradley-Terry Reward Model and Preference Model

**Bradley-Terry model construction.** We follow the previous works (Ouyang et al., 2022; Bai et al., 2022a) to initialize the reward model using the SFT model[2]. We replace the last layer with a linear head to predict a scalar score suitable for preference learning. The reward model is trained using the negative log-likelihood loss function, enabling maximum likelihood estimation (MLE). This loss function is defined as:

$$L_{\mathrm{RM}}(\theta) = -\mathbb{E}_{x,a^w,a^l \sim \mathcal{D}} \log \sigma\big(r_\theta(x, a^w) - r_\theta(x, a^l)\big),$$

where $a^w$ is the preferred response over $a^l$. We train the LLaMA-3-8B-based reward model for one epoch with a global batch size of 512. The learning rate is set to lr $= 2 \times 10^{-6}$, and a cosine learning rate schedule with a warm-up ratio of 0.03 is employed.

---

[2]For preference/reward modeling, we use meta-llama/Meta-Llama-3-8B-Instruct since only this checkpoint is available in the early stage of this project.

Table 1: Comparison of the test accuracy between the Bradley-Terry (BT) reward model and the preference model. We evaluate the model using the Reward-Bench (Lambert et al., 2024).

| Base Model | Type | Data Mixture | Chat | Chat Hard | Safety | Reasoning |
|---|---|---|---|---|---|---|
| LLaMA-3-8B-it | Prompting | - | 93.6 | 44.3 | 71.3 | 73.5 |
| LLaMA-2-13B | BT | mix1 | 96.4 | 55.5 | 55.0 | 62.4 |
| LLaMA-3-8B-it | BT | mix2 | **99.4** | 65.1 | 87.8 | 86.4 |
| LLaMA-3-8B-it | Preference | mix2 | 98.3 | **65.8** | **89.7** | **94.7** |

**Preference model construction.** A (pairwise) preference model takes a prompt $x$ and two responses $a^1, a^2$ as the input and predicts the probability of $\hat{\mathbb{P}}(a^1 \succ a^2 | x, a^1, a^2)$ (Jiang et al., 2023). We follow Zhao et al. (2023); Liu et al. (2023a) to leverage the LLM's capability as a next-token predictor for preference modeling. Specifically, for a given preference pair $(x, a^1, a^2, A)$, where $A$ indicates that the first response is preferred, the pair is formatted as an instruction-following task:

$$\text{instruction} = [\text{CONTEXT}]\ \{x\}\ [\text{RESPONSE A}]\ \{a^1\}\ [\text{RESPONSE B}]\ \{a^2\}, \text{ and label} = A.$$

If the second response is preferred, we replace the label A with B. Then, we simply treat the preference modeling as an instruction-following task to fine-tune the model on these instruction-label pairs. To mitigate position bias (the preference model may prefer the response that is given in the position of RESPONSE A), the order of the responses is randomized during data formatting. During inference, if we denote the probability of decoding A as $p_A$ and the probability of decoding B as $p_B$, then $\hat{\mathbb{P}}(a^1 \succ a^2 | x, a^1, a^2)$ is taken as $p_A/(p_A + p_B)$. We train the LLaMA-3-8B-based preference model for one epoch. The samples are packed into blocks with length 3072 and a global batch size of 128 is used. The learning rate is set to lr $= 5 \times 10^{-6}$, and a cosine learning rate schedule with a warm-up ratio of 0.03 is employed. We mention in passing that it is possible to include detailed rubrics in the data format to further improve the preference dataset, which we leave for future work (Qin et al., 2023).

## 2.2 Evaluation Result

We evaluate the models using the RewardBench (Lambert et al., 2024), a benchmark designed to assess reward model capabilities in four categories: Chat, Chat-Hard, Safety, and Reasoning. The main evaluation results are in Table 1. It is evident that without explicit training, the prompting approach is inferior to both the BT model and the preference model across all metrics. Meanwhile, the preference model outperforms the BT model in reasoning tasks related to coding and math. We also notice that with more data, especially data specified in coding, math, and safety, the reward model trained by mix2 achieves higher accuracy in safety and reasoning compared with early versions of attempts. In particular, we use the Ultra-RM-13B as a reference model in Table 1, where we can observe that the extra data related to safety and reasoning (as well as a stronger base model) largely contribute to the superior performance of our reward model.

**Length bias in reward modeling.** It is known that the LLMs aligned by RLHF usually give longer responses (Xiong et al., 2023; Yuan et al., 2024b;b), where the length bias also exists in the reward models, likely influenced by the preference data used. To better understand this bias, we randomly sample 2K prompts from the prompt set and use the SFT model to generate 8 responses per prompt. Then, we compute the lengths and rewards of the responses and plot the heatmaps of the Pearson correlation coefficient between them in Figure 3. Clearly, both of the two reward models are biased toward the longer responses to some degree. In comparison, UltraRM-13B demonstrates a stronger bias, as we observe that the mean coefficient of the UltraRM-13B (left) is 0.19, while it is 0.06 for our BT reward (right). This may partially result from the use of additional Capybara, OpenOrca, and UltraInteract, whose preferred responses are shorter than the rejected responses. We will return to the ablation study of the impacts of the reward models in Section 4.

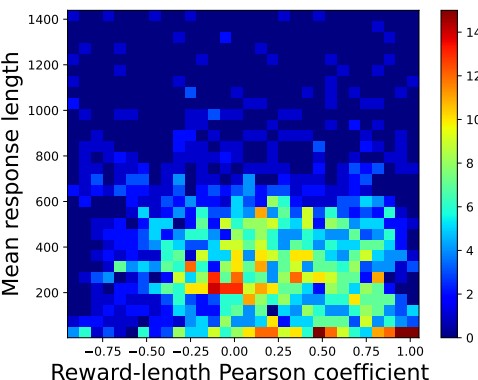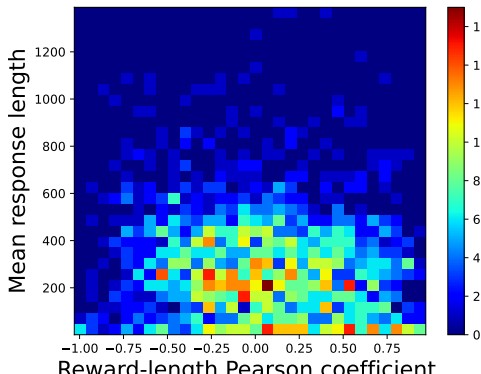

Figure 3: The heatmap of the Pearson correlation coefficients between reward and response length. For each prompt, we use the SFT model to generate 16 responses and compute the coefficient. We also group the prompts by the mean responses (y-axis). The left figure is the UltraRM-13B and the right one is our BT reward based on mix2. We observe that the heatmaps for both UltraRM-13B and our BT reward model show a tendency toward the positive side. This indicates that the reward models exhibit a preference for longer responses.

## 3 Iterative Policy Optimization

We develop the main algorithms for the online iterative RLHF in this section, with both theoretical insights and implementation details. In particular, the algorithms are in a direct preference learning style for stable and efficient training.

### 3.1 Supervised Fine-Tuning

The base model used in this project is LLaMA-3-8B. To ensure the reproducibility and openness of the project, we perform SFT by ourselves to obtain the initial policy $\pi_0$. We collect a set of high-quality instruction datasets for SFT, such as ShareGPT (Chiang et al., 2023), SlimOrca (Lian et al., 2023b), MathInstruct (Yue et al., 2023), and Evol-Instruct (Xu et al., 2023a) (see the Appendix for a full list). The training is carried out for one epoch with a learning rate of $2 \times 10^{-5}$. A cosine scheduler is employed, and the global batch size is set to 32 with a warm-up ratio of 0.03. To accelerate training, we follow Diao et al. (2023); Tunstall et al. (2023) to pack the samples and use a block size of 8192.

### 3.2 Iterative Direct Preference Learning: Theoretical Insights and Algorithmic Principles

From a theoretical perspective, one of the most critical metrics in the reinforcement learning process is the uncertainty estimator $\Gamma$. For example, considering the linear case, $r = \langle \theta, \phi(x, a) \rangle$ such that $\theta \in \mathbb{R}^d, \|\theta\| \leq B$. For any two policies $\pi_t^1, \pi_t^2$, we define the information gain as $\Gamma_t(\lambda, \pi_t^1, \pi_t^2) = \beta\|\mathbb{E}_{\pi_t^1}\phi(x, a_t^1) - \mathbb{E}_{\pi_t^2}\phi(x, a_t^2)\|_{\Sigma_t^{-1}}$ ($\beta$ is a constant coefficient), which is the projection of the **new feature difference** to the **historical feature covariance matrix** $\Sigma_t = \lambda I + \sum_{s=1}^{t-1} \mathbb{E}_{x \sim d_0, a^1 \sim \pi_s^1, a^2 \sim \pi_s^2} \left[ (\phi(x, a^1) - \phi(x, a^2))(\phi(x, a^1) - \phi(x, a^2))^\top \right]$.

We present the main algorithmic framework in Algorithm 1 for the online iterative RLHF, and summarize the key principles and algorithmic ideas as follows.

**Hybrid batch learning.** We formulate a slightly more general framework to combine an initial offline dataset with online data collected during training, hence the name hybrid learning, similar to the recipe of Claude (Bai et al., 2022a) and LLaMA-2 (Touvron et al., 2023). We also use a large batch size m for sparse updates.

---

**Algorithm 1** Theoretical Online Iterative RLHF with Enhancer

---

1: **Input:** offline dataset $\mathcal{D}_{\text{off}}$ (can be empty); batch size $m > 0$, and preference model $\mathbb{P}$, the number of iteration T.
2: **for** t=1,...,T **do**
3:   Exploitation with the main agent: denote the MLE $r_{\text{MLE}}$ (no need to explicitly compute the reward function if we use DPO) and compute the best guess we have so far:

$$\pi_t^1 = \underset{\pi \in \Pi}{\arg\max} \, \mathbb{E}_{x \sim d_0} \mathbb{E}_{a \sim \pi(\cdot|x)} \left[ r_{\text{MLE}}(x, a) - \eta D_{\text{KL}}(\pi(\cdot|x) \| \pi_0(\cdot|x)) \right]. \quad (7)$$

4:   Exploration with the enhancer: given the policy, assume we have the uncertainty quantifier with respect to $\pi_t^1$ as $\Gamma_t^m(\lambda, \pi_t^1, \pi^2)$ and compute the policy $\pi_t^2 = \arg\max_{\pi^2 \in \Pi_t} \Gamma_t^m(\lambda, \pi_t^1, \pi^2)$ where

$$\Pi_t = \{\pi' \in \Pi : \underbrace{\eta \mathbb{E}_{x \sim d_0} D_{\text{KL}}(\pi(\cdot|x), \pi^1(\cdot|x))}_{\text{How far does the enhancer move away.}} \leq \underbrace{\Gamma_t^m(\lambda, \pi_t^1, \pi')}_{\text{How much information we can get.}} \}. \quad (8)$$

5:   Collect $\mathcal{D}_t = \{(x_i, a_i^1, a_i^2, y_i)\}_{i=1}^m$ by $x_i \sim d_0, a_i^1 \sim \pi_t^1(\cdot|x_i), a_i^2 \sim \pi_t^2(\cdot|x_i)$ and $y_i \sim \text{Ber}\left(\mathbb{P}(a_i^1 \succ a_i^2 | x, a_i^1, a_i^2)\right)$;
6: **end for**
7: **Output:** the best policy in $(\pi_{1:T}^1)$ by a validation set.

---

**Non-symmetric structure to balance *exploitation* and *exploration*.** The framework also features a non-symmetric structure by involving a main agent and an enhancer.

- **Main agent aims to learn $\pi^*$.** Specifically, for each iteration, the first agent, referred to as the main agent, always takes the optimal policy under the MLE $r_{\text{MLE}}$ of the historical data, which can be viewed as a fully **exploitation** of the information we collected so far;

- **Enhancer aims to assist the main agent's learning.** Since the main agent solely exploits the historical data, it is effective only when we can continuously obtain new information about the alignment problem from the newly collected online data or the offline data $\mathcal{D}_{\text{off}}$ has provided enough coverage (which is unlikely to hold in practice as we discuss in Section 1.1). The enhancer, therefore, **explores** in the direction where there is more uncertainty relative to the main agent's policy $\pi_t^1$ (measured by $\Gamma_t^m(\lambda, \pi_t^1, \pi')$), while maintaining a moderate KL divergence with $\pi_t^1$.

We have the following theoretical guarantees when strategic exploration methods are applied.

**Theorem 1** (Informal Theorem 2 in (Xiong et al., 2023)). *For any precision parameter $\epsilon > 0$, with a batch size $m = \widetilde{O}(\frac{d_e}{\epsilon^2})$ and other suitable choices of hyper-parameters, then, with high probability, after at most $T = \widetilde{O}(d_e)$ iterations, we can find a $t_0 \in [T]$ so that $J(\pi^*) - J(\pi_{t_0}) + \eta \mathbb{E}_{x_{t_0} \sim d_0} \left[ D_{\text{KL}}(\pi^*(\cdot|x_{t_0}) \| \pi_{t_0}(\cdot|x_{t_0})) \right] \lesssim \epsilon$, where $J(\pi) = \mathbb{E}_{x \sim d_0} [\mathbb{E}_{a \sim \pi(\cdot|x)}[r^*(x, a)] - \eta D_{\text{KL}}(\pi(\cdot|x) \| \pi_0(\cdot|x))]$ is the KL-regularized value. Here $\widetilde{O}$ hides some log factors and $d_e$ is a complexity measure of the RLHF problem. In particular, if the reward function can be embedded into a $d$-dimensional space that is linear in the feature map of $\phi : \mathcal{X} \times \mathcal{A} \to \mathbb{R}^d$, we have $d_e = d$. (Zhong et al., 2022; Liu et al., 2023b)*

### 3.3 Practical Implementation Details

We now shift our focus from the theoretical insight to the practical implementation. We provide an illustration of our implementation in Fig. 4.

**The MLE policy.** Since the main agent only exploits the data, we can run DPO on the historical data to approximate the optimal policy under the $r_{\text{MLE}}$: $\pi_t^{\text{MLE}}$. We remark that while we use DPO here due to its simplicity, the Algorithm 1 can be implemented by combining it with any oracle algorithms (e.g., PPO and InfoNCA (Chen et al., 2024a)) that are approximations of the KL-regularized optimization problem.

**Exploration policy.** The primary challenge lies in the choice of enhancer policy for exploration. Recall that our goal is to find an enhancer policy that maximizes the relative uncertainty to the main agent from the

---

**Algorithm 2** Practical Version of Online Iterative RLHF with BT Reward Model

---

1: **Input:** offline dataset $\mathcal{D}_{\text{off}}$ (can be empty); batch size $m > 0$, rejection sampling parameter $n$ and reward model $r$; the number of iteration T.
2: **for** t=1,...,T **do**
3:     Compute $\pi_t$ by the DPO algorithm with $\mathcal{D}_{\text{off}} \cup \mathcal{D}_{1:t-1}$ using the SFT policy $\pi_0$ as reference model.
4:     Sample a batch of prompts $\{x_i\}_{i=1}^m$ from $d_0$. For each prompt, we sample $n/2$ responses using $\pi_t$ with temperature 1.0 and $n/2$ responses using $\pi_t$ with temperature 0.7.
5:     For each prompt $x_i$, we rank them using $r$ and take the best response and the worst one to construct a preference pair into $\mathcal{D}_t$. Eventually, we collect $m$ preference pairs.
6: **end for**
7: **Output:** the best policy in $(\pi_{1:T}^1)$ by a validation set.

---

confidence set defined in Equation (8). Unfortunately, the uncertainty estimator does not have an analytical form except the linear case. But the main insight we can derive here is to maximize the policy difference with $\pi_t^1$, while maintaining a moderate KL divergence. This motivates us to use model variants of $\pi_t^1$. We discuss some popular heuristic implementations here.

- **Adjusting Temperature and Training Steps.** In the project of Claude (Bai et al., 2022a), the authors choose to use the models with different training steps as $(\pi_t^1, \pi_t^2)$. For instance, if we run PPO for 2 epoch in total, we may take $\pi_t^1$ as the model saved at the end of the first epoch and take $\pi_t^2$ as the one saved at the end of the second epoch. Additionally, the LLaMA-2 project (Touvron et al., 2023) adjusts the sampling temperature of $\pi_t^1$ to induce $\pi_t^2$. These modifications introduce diversity in the models and facilitate exploration.

- **Rejection Sampling** is another popular ensemble-based exploration approach (Nakano et al., 2021; Dong et al., 2023; Gulcehre et al., 2023). In the context of LLMs, it is typically restricted to the best-of-$n$ sampling. Specifically, we sample $n$ independent responses by $\pi_t^1$ for each prompt, and then use a preference/reward function to rank the responses and take the one with the highest reward as the final output. In other words, we take $\pi_t^2$ as the best-of-n variant of $\pi_t^1$. In this way, the $\pi_t^2$ enlarges the margins between $\pi_t^1$ and provides exploration. Meanwhile, in this case, the KL divergence between the two policies is upper bounded by $\log n - \frac{n-1}{n}$ and is usually far better than this conservative estimation (Beirami et al., 2024);

- After the first submission of this work, there is a line of works proposing to use biased DPO loss in the online iterative framework to encourage exploration (Xie et al., 2024; Zhang et al., 2024; Cen et al., 2024), whose idea originates from the theoretical RL study (Xiong, 2023; Liu et al., 2023b). Specifically, they add a SFT loss term, also known as the "feel-good" term, into the loss function so that the algorithm favors the model that is more optimistic. We refer the interested readers to these works for a more detailed algorithm description and empirical results. We also integrate the option of adding such a bias term in the revision of our public code.

In our experiments, we use the DPO to approximate the computational oracle and implement DPO with the open-source package TRL[3]. We run DPO with the reference model $\pi_0$ (the SFT model) on the historical data for 2 epochs to get the MLE policy $\pi_t^{\text{MLE}}$. We use a cosine learning rate scheduler with a peak learning rate of 5e-7 and 0.03 warm-up ratio. We use a global batch size of 128 and use a KL coefficient of $\eta = 0.1$. To accelerate training, we do not restart from $\pi_0$ at each iteration as in Bai et al. (2022a); Xiong et al. (2023) but use the last-iteration model as the initial checkpoint and use $\pi_0$ as the reference model. In this way, the data used for training is the same as that of Bai et al. (2022a) and Xiong et al. (2023) but is of a different order. We do not see performance regression with this choice, and it saves us for half of the training time.

To facilitate exploration, we combine the temperature tuning with the rejection sampling strategy with $n = 8$. Instead of fixing $\pi_t^1 = \pi_t^{\text{MLE}}$ (like the center of confidence set) and optimizing the $\pi_t^2$ solely to be the best-of-8 variant of $\pi_t^{\text{MLE}}$, we take $\pi_t^1$ and $\pi_t^2$ as the best-of-8 policy and worst-of-8 policy induced by $\pi_t^{\text{MLE}}$.

---

[3]https://github.com/huggingface/trl

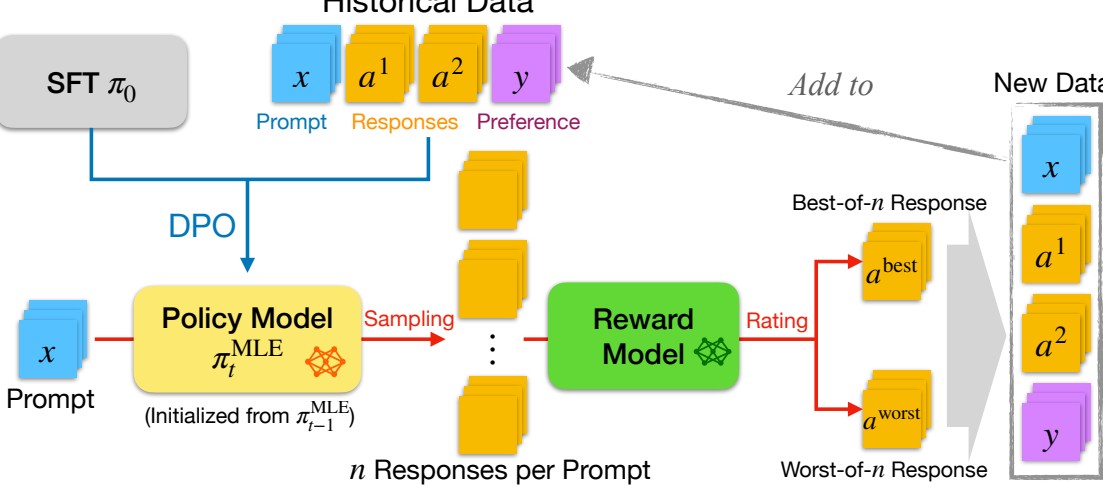

Figure 4: Illustration of our implementation of iterative direct preference learning. In iteration $t = 1$, the historical dataset is empty, and the resulting policy model $\pi_1^{\mathrm{MLE}}$ is the same as its initialization, $\pi_0$, which is the SFT model checkpoint. After that, the historical dataset grows with preference data collected from previous iterations.

In other words, we take the best response and the worst response as ranked by the reward model to get a preference pair. In this case, we jointly optimize the two policies to maximize their difference (measured by the uncertainty), which tends to be more efficient in practice and enjoys the same theoretical guarantee as stated in Theorem 1. This choice is similar to Hoang Tran (2024); Pace et al. (2024); Yuan et al. (2024b); Xu et al. (2024). We also drop the pair where $\pi_t^1$ and $\pi_t^2$ give the same response, which implies that the uncertainty in this direction is already small. For this round of experiments, we still use the reward function trained as the MLE of the BT reward model to rank the responses for the following reasons. First, to rank $n$ responses, the complexity of using the reward model is linear in $n$, while it is far more complicated with the pairwise preference model. Second, during the early experiments, we observe significant length bias in the iterative RLHF. Therefore, we would like to explore the strategy to mitigate the length bias, and it is relatively easier to penalize the reward value with the length of the response. Finally, the BT reward model is comparable with the preference model except for the reasoning task and it may be already satisfactory for our goal. We leave a more comprehensive comparison between the BT reward model and preference model for future study.

**Prompt set, and data generation.** We collect prompts from UltraFeedback (Cui et al., 2023), HelpSteer (Wang et al., 2023), OpenOrca (Lian et al., 2023a), UltraInteract (Yuan et al., 2024a), Capybara (Daniele & Suphavadeeprasit, 2023) and DIBT-10K[4] and prepare the full prompt set.In our experiments, we use a subset of 60K prompts and iterate for three iterations, so 20K prompts are used to generate 20K x 16 responses per iteration. To accelerate data generation, we use VLLM (Kwon et al., 2023) for inference. We set the max generation length as 2048, and use a sampling temperature of 1.0/0.7 without any top-k/top-p strategy. To offer a more intuitive comprehension of our prompts collection, we provide visualization plots in the Appendix (Figure 5).

---

[4]https://huggingface.co/datasets/DIBT/10k_prompts_ranked

Table 2: Evaluation results and comparison between the resulting models and existing models. * means that the model is based on the mixture-of-experts architecture. We report the length-control win rate of AlpacaEval-2 as recommended by the authors. RS is short for rejection sampling (Dong et al., 2023) and X means that the value is unavailable. Only underline results are better than our 8B model.

| Model | Size | Method | LC AlpacaEval-2 | MT-Bench | Chat-Arena-Hard |
|---|---|---|---|---|---|
| Gemma-7B-it | 7B | SFT | 10.4 | 6.38 | 7.5 |
| Zephyr-7B-beta | 7B | Vanilla DPO | 13.1 | 7.34 | X |
| Mistral-7B-v0.2-it | 7B | SFT | 17.1 | 7.51 | 12.6 |
| Open-Chat-0106 | 7B | SFT | 15.6 | 7.8 | X |
| Starling-7B-beta | 7B | PPO | 25.8 | 8.12 | 23.0 |
| LLaMA-3-8B-it | 8B | RS+DPO+PPO | 22.9 | 8.16 | 20.6 |
| Ours (SFT baseline) | 8B | SFT | 10.2 | 7.69 | 5.6 |
| Ours (DPO baseline) | 8B | Vanilla DPO | 22.5 | 8.17 | 22.4 |
| Ours (Iterative RLHF) | 8B | Iterative DPO | **31.3** | **8.46** | **29.1** |
| Vicuna-33b-v1.3 | 33B | SFT | 17.6 | 7.12 | 8.6 |
| Yi-34B-Chat | 34B | SFT | 27.2 | X | 23.1 |
| Mixtral-8x7B-it | 45B* | SFT | 23.7 | 8.30 | 23.4 |
| Tulu-2-DPO-70B | 70B | Vanilla DPO | 21.2 | 7.89 | 15.0 |
| LLaMA-3-70B-it | 70B | RS+DPO+PPO | 34.4 | 8.95 | 41.1 |
| Mixtral-8x22B-it | 141B* | SFT | 30.9 | 8.66 | 36.4 |
| GPT-3.5-turbo-1106 | - | - | 19.3 | 8.35 | 18.9 |
| GPT-3.5-turbo-0613 | - | - | 22.7 | 8.39 | 24.8 |
| GPT-4-0613 | - | - | 30.2 | 9.18 | 37.9 |
| Claude-3-Opus | - | - | 40.5 | 9.00 | 60.4 |
| GPT-4 Turbo (04/09) | - | - | 55.0 | X | 82.6 |

## 4 Evaluation of the Model

### 4.1 Benchmarks

We evaluate the models by standard benchmarks, including AlpacaEval-2, MT-Bench, and Chat-Arena-Hard. Details are provided in the Appendix.

We also measure the ability of the resulting models using academic benchmark, including GSM-8K (Cobbe et al., 2021), MMLU (Hendrycks et al., 2020), HumanEval (Chen et al., 2021), TruthfulQA (Lin et al., 2021), ARC (Clark et al., 2018), and MBPP (Austin et al., 2021). These benchmarks evaluate the models' ability in coding, reasoning, and general knowledge. In particular, it is known that RLHF alignment can introduce performance degeneration in reasoning, calibration (providing accurate confidence estimates), and truthfulness capabilities (generating accurate and factual responses), which is also referred to as the alignment tax in the literature (Ouyang et al., 2022; Bai et al., 2022a; OpenAI, 2023). Therefore, evaluating our model on these benchmarks is crucial to understanding the impact of iterative RLHF on these specific aspects.

### 4.2 Main Results

**Online iterative RLHF significantly improves conversation quality.** We evaluate our model's conversation abilities using AlpacaEval-2, MT-Bench, and Chat-Arena-Hard, (results in Table 2). Compared to other open-source models with less than 10B parameters, our model outperforms them on the conversation and instruction-following benchmarks with a significant margin. Notably, our model trained with iterative DPO consistently outperforms that of vanilla offline DPO (`DPO baseline`). This demonstrates the advantage of online iterative RLHF. Moreover, our model outperforms the Tulu-2-DPO-70B and GPT-3.5-turbo-1106, which are aligned by DPO or PPO and are much larger than our base model. These results show that the

Table 3: Evaluation results of the resulting model on academic benchmarks and comparison with other open-access LLMs.

| Model | Size | Method | GSM-8K | MMLU | HumanEval | TruthfulQA | ARC | MBPP |
|---|---|---|---|---|---|---|---|---|
| LLaMA-3-8B-it | 8B | RS+DPO+PPO | 79.6 | 66.0 | 61.6 | 43.9 | 59.5 | 61.1 |
| Ours (SFT baseline) | 8B | SFT | 74.2 | 64.7 | 65.2 | 53.4 | 61.4 | 62.3 |
| Ours (DPO baseline) | 8B | Vanilla DPO | 79.8 | 64.5 | 63.4 | 61.8 | 65.2 | 60.3 |
| Ours (Iterative RLHF) | 8B | Iterative DPO | 80.7 | 65.3 | 64.6 | 60.4 | 64.3 | 60.8 |

online iterative RLHF can effectively adjust the style of the model responses, thus improving the conversation quality.

**Academic Task.** As RLHF can impact a model's reasoning and calibration abilities, typically in a negative way (Bai et al., 2022a; Ouyang et al., 2022; OpenAI, 2023), we compare our model's performance on academic benchmarks (Table 3) with the SFT checkpoint and other baselines. We don't observe significant performance regression compared to the SFT baseline. Interestingly, our iteratively DPO-aligned model even outperforms the SFT model in GSM-8K, MMLU, TruthfulQA, and ARC benchmarks. We believe that these increased capacities of the model are injected in the pre-training stage and SFT stage, and iterative DPO helps it leverage them more effectively. This is because the 60K alignment data used in the iterative RLHF are orders of magnitude less than those used in the previous two stages.

**Remark 1.** *The RLHF-aligned models based on some initial checkpoints and more epochs can approach or even outperform the state-of-the-art closed-source models like GPT-4 and Claude on benchmarks. However, we remark that we should be more careful in interpreting these results because the test sets of the benchmarks are finite and may not be representative enough to capture the complicated real-world scenarios. Moreover, the increased possibility of small models overfitting the benchmarks may lead to benchmark hacking, which means that the real capacity of a model with a high score is still limited. In particular, while it is possible to get even higher results on the benchmark (e.g., 44.84 in LC AlpacaEval-2 and 35.7 in Chat-Arena-hard, but the performance on academic benchmarks drops significantly), we presented our current model by human evaluation on some randomly chosen test prompts. We also found that GPT-based evaluation highly depends on the configuration. Our model obtains 37.2 LC win-rate (45.4 win-rate) with "alpaca_eval_gpt4_turbo_fn" config, which has better agreement with human evaluation.*

**Ablation study on filtering data with length penalty.** We observed that the aligned model's response length was significantly longer than the SFT baseline (potentially due to reward model bias as shown in Figure 3). To address this, we conducted an ablation study by incorporating a length penalty into the reward function:

$$\widetilde{r}(x, a) = \hat{r}(x, a) - \lambda|a|, \tag{9}$$

where $|a|$ is the number of **characters** of the response. We compare the model trained with this penalty to the vanilla version and report the results in Table 4. As expected, the length penalty effectively mitigated the length bias, leading to shorter responses. In particular, the model trained with length penalty achieves a superior *length-control* AlpacaEval-2 win rate, as well as better results on some academic benchmarks. This demonstrates the advantage of mitigating length bias and motivates us to study the verbosity issue in reward modeling further. Finally, we notice that the model trained with length penalty is worse in the Chat-Arena-Hard benchmark. This may suggest that we also need a length-control version for this benchmark to provide a more reasonable evaluation.

Table 4: Ablation study on the impact of reward models and length penalty in the online iterative RLHF. The response length is averaged over the responses to the Chat-Arena-Hard Benchmark.

| RM/Model | Len. Pen. | LC Alp. | Arena-H. | Len. | GSM-8K | MMLU | HumanEval | TruthfulQA | ARC | MBPP |
|---|---|---|---|---|---|---|---|---|---|---|
| Ours | - | 31.3 | 29.1 | 656 | 80.7 | 65.3 | 64.6 | 62.2 | 64.3 | 60.8 |
| Ours-concise | 0.001 | 38.1 | 22.1 | 382 | 78.8 | 65.5 | 66.5 | 60.4 | 65.1 | 62.4 |
| UltraRM-13B | - | 20.7 | 24.3 | 745 | 78.9 | 64.9 | 63.7 | 59.9 | 63.6 | 60.8 |

**On the impact of reward model.** We investigate the effects of the reward (preference) model used in the online iterative RLHF. Our model's performance is compared to a model trained with UltraRM-13B, and the ablation study results are summarized in Table 4. We observe that the model trained with UltraRM-13B has longer responses than ours, which is consistent with its stronger bias, as shown in Figure 3. Considering the alignment tax, the accuracy on the academic benchmarks drops more than our models. One important reason is that the UltraRM-13B does not have a good reasoning ability (see Table 1), so it may not provide appropriate preference signals for reasoning-related conversions. For instance, the model may favor some responses with many comments in the coding task, which tend to be very helpful but are indeed useless when evaluated by humans. Notably, the model trained with UltraRM-13B achieves a higher Chat-Arena-Hard win rate than our concise version, which also supports the verbosity bias of the Arena-Hard benchmark. During the training process, we also observe that the model trained with UltraRM-13B achieves a lower training loss, which may suggest that the signals of UltraRM-13B are more consistent and easy to learn. In contrast, the convergence under our reward model is slower due to the complex preference signal.

## 5 End Note and Future Direction

In this technical report, we study the workflow of the online iterative RLHF, which leverages on-policy sampling and external preference signals from a proxy preference model trained on a diverse set of open-source preference datasets. The resulting model demonstrates impressive performance on standard benchmarks, and the report provides detailed instructions for reproducing the results, including data, code, models, and hyper-parameter choices.

There are still many potential directions to explore. First, as we can see in Table 4, the iterative RLHF heavily relies on the quality of the preference signal. In this project, we use a proxy scalar reward model trained on a diverse set of open-source datasets to approximate human feedback. It would be interesting to see whether we can design a more effective strategy to model different types of preference signals, like a multi-head reward (Wang et al., 2024) and classification-based activation strategy (Touvron et al., 2023). Second, while the rejection sampling seems to be a good heuristic exploration strategy, it is still interesting to see whether we can design more effective ways for exploration. Finally, most of the models after RLHF tend to reply the prompts with much longer responses. Such a length bias is further amplified in the iterative RLHF framework. We presented a preliminary study on this issue by leveraging an additional length penalty in reward for data filtering. It would be interesting to see whether we can further mitigate this issue by additional algorithmic designs or post-training techniques.

We hope the results of this project can advance the direction of online iterative RLHF and contribute to the training of stronger and larger open-source LLMs.

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

# A  Authorship and Credit Attribution

All authors provided valuable contributions to this project, each bringing unique expertise and insights that were crucial for its success.

**HD** first demonstrated that iterative DPO algorithm can achieve state-of-the-art performance; wrote a development version code for SFT, iterative RLHF; contributed to the training of SFT model and BT-RM; conducted extensive experiments on training and hyper-parameter tuning of iterative RLHF; delivered the released BT reward model; provide preference dataset for final BT-RM and some initial versions of the prompt data; conducted the RM evaluation and GPT-based evaluation of the generative models; contributed to paper writing; contributed to the public version of iterative RLHF code.

**WX** wrote the codes for the Bradley Terry reward model and conducted most of the experiments for both reward and preference model training; delivered the released pairwise preference model; contributed to the preference dataset search and hyper-parameter tuning; initiated and organized the online iterative RLHF project; wrote the initial code for the online iterative DPO and prepared its public version on GitHub; contributed to the evaluation of the reward and preference models; assisted in the collection and the cleaning of the preference dataset; wrote the paper.

**BP** conducted most of the final SFT and RLHF experiments and delivered the released SFT and RLHF model; independently wrote a development version code for SFT and iterative RLHF; developed the SFT recipes (data, hyper-parameter, model selection); conducted extensive experiments on the training and hyper-parameter tuning of SFT, offline and iterative RLHF; conducted the GPT-based evaluation and all the academic benchmarks; contributed to paper writing.

**HW** initiated the training code of the pairwise preference model, and conducted experiments in the training of the Bradley Terry reward model and pairwise preference model; collected, filtered, and deduplicated the prompt set; contributed to the preference dataset collection and data cleaning; contributed to the evaluation and analysis of the reward and preference models; made substantial writing contributions to the reward modeling section and created illustrative figures for the algorithmic frameworks and dataset visualization.

**HZ, YZ, NJ, DS, CX, TZ** supported and advised the works of the junior authors, provided computational resources, and suggested experiments and writings.

# B  Additional Experimental Details

## B.1  Preference Datasets

Following the Pair-RM (Jiang et al., 2023) and LLaMA-2 (Touvron et al., 2023), we use a mixture of open-source datasets as the training set. Here is a brief introduction to the datasets:

- **HH-RLHF** (Bai et al., 2022a) is a pairwise preference dataset where each sample is accompanied by a conversation history and two alternative responses written by an early Claude model with 52B parameters. The preferences of the responses are annotated by humans.

- **SHP** (Ethayarajh et al., 2022) is sourced from Reddit and includes examples from 18 subreddits, such as askacademia, askbaking, askengineers, and changemyview. Each example is a Reddit post with a question/instruction and a pair of top-level comments. One comment is preferred by more Reddit users than the other. All preferences and responses are provided by humans. Following Cui et al. (2023), only samples with a score ratio > 2 are used, and at most 5 pairs are taken for each prompt.

- **HelpSteer** (Wang et al., 2023). This open-source dataset (Wang et al., 2023) contains prompts, responses, and five human-annotated attributes (helpfulness, correctness, coherence, complexity, and verbosity) ranging from 0 to 4. The prompts are generated using a mixture of template-generated and human-generated methods, while responses are generated by an in-house LLM. The authors generate up to 4 responses per prompt, and we can construct pairwise comparisons based on them.

| Dataset | #Prompts | Prompt Len. | Preferred Len. | Rejected Len. | Completion | Annotator | #Pairs |
|---|---|---|---|---|---|---|---|
| HH-RLHF | 115092 | 160.4 | 82.2 | 73.6 | LLM | Human | 115396 |
| SHP | 31003 | 186.2 | 173.6 | 88.8 | Human | Human | 93301 |
| HelpSteer | 8592 | 530 | 116.4 | 89.3 | LLM | Human | 37131 |
| PKU-SafeRLHF-30K | 6975 | 21.5 | 70.4 | 74.6 | LLM | Human | 26874 |
| UltraFeedback | 63591 | 161.5 | 279.5 | 211.1 | LLM | GPT-4 | 340025 |
| UltraInteract | 76086 | 507.4 | 396.6 | 416.7 | LLM | GPT-4 | 161927 |
| CodeUltraFeedback | 9938 | 172.8 | 427.6 | 400.6 | LLM | GPT-3.5 | 50156 |
| Argilla-Math | 2352 | 36.5 | 276.5 | 265.3 | LLM | GPT-4 | 2418 |
| OpenOrca | 6791 | 153.3 | 165.4 | 260.5 | LLM | GPT-4 | 6926 |
| Capybara | 14740 | 634.5 | 348.4 | 401.9 | LLM | GPT-4 | 14811 |

Table 5: A summarization of open-source preference datasets. "Prompt Len." represents the average prompt length in terms of tokens, and "Preferred/Rejected Len." stands for the average length of preferred or rejected responses. All of these lengths are averaged over all pairs and we use the tokenizer of LLaMA-3-8B. "Completion" marks the data source of the text completions for prompts. We apply pre-processing techniques to all these datasets and delete the noisy samples from the original dataset. For the dataset whose prompt is with multiple responses, we include all the possible comparisons except those with the same score/ranking to compute the total number of comparison pairs.

- **PKU-SafeRLHF** (Ji et al., 2024). This dataset (Ji et al., 2024) consists of 30k+ expert comparison data. Each sample includes two responses to a question and two preference signals for helpfulness and safety, respectively. The responses are generated by open-source chatbots, and the preference signals are merged through the results of 14 harm category multi-class classficiation.

- **UltraFeedback** (Cui et al., 2023) consists of 64k prompts from diverse resources (including UltraChat, ShareGPT, Evol-Instruct, TruthfulQA, FalseQA, and FLAN) and the authors generate 4 responses per prompt using 4 different LLMs sampled from a diverse set of state-of-the-art open-source LLMs. The preference is from GPT-4 based on a fine-grained annotation instruction, which contains 4 different aspects, namely instruction-following, truthfulness, honesty and helpfulness. The dataset collection strategy of UltraFeedback has also influenced many subsequent works.

- **CodeUltraFeedback** is generated similarly with the Ultrafeedback but focuses on the coding task. The annotation is from GPT-3.5.

- **UltraInteract** (Yuan et al., 2024a) is a preference dataset designed for complex reasoning tasks. The authors collect a preference tree for each instruction, with the instruction being the root and each action a node. A trajectory is a root-to-leaf path consisting of a sequence of actions. Paired correct and incorrect nodes or trajectories are used for preference learning.

- **Distilabel-Capybara**[5] is a preference dataset of multi-turn dialogues whose prompts are taken from Daniele & Suphavadeeprasit (2023), where the responses are generated by open-source LLMs and preferences are generated by GPT-4.

- **Distilabel-Orca**[6] is collected similarly with Capybara but with the prompts from Lian et al. (2023a).

The training of LLMs is highly data-dependent. To ensure high-quality training data, we conduct a filtering process on the open-source datasets we use. This process removes low-quality and meaningless samples. Additionally, conversations with empty rounds or incorrect labels (implied by the other features of the dataset) are eliminated. Furthermore, in datasets where absolute scores are available, pairwise comparisons with small margins are excluded as these preference signals tend to be noisy (Bansal et al., 2023). This process roughly deletes 10% of the data. We summarize the statistics of the open-source datasets that are used for the training in Table 5 and prepare them, as well as our data filtering script, on the huggingface.

---

[5]https://huggingface.co/datasets/argilla/distilabel-capybara-dpo-7k-binarized
[6]https://huggingface.co/datasets/argilla/distilabel-intel-orca-dpo-pairs

Table 6: A summarization of the benchmarks we use in this project. We list the metric and number of shots (indicating zero-shot learning or in-context learning) used for LLM evaluation on each dataset.

| Benchmark | LC-AlpacaEval-2 | MT-Bench | Chat-Arena-Hard | GSM-8K | MMLU | HumanEval | TruthfulQA | ARC | MBPP |
|---|---|---|---|---|---|---|---|---|---|
| Metric | win rate | score | win rate | acc | acc | acc | acc | acc | acc |
| Num. of Shots | 0 | 0 | 0 | 8 | 5 | 0 | 0 | 25 | 0 |

We consider two versions of the training set:

- MIX1: HH-RLHF + SHP + UltraFeedback + Summarization (Stiennon et al., 2020).

- MIX2: all the datasets in Table 5.

The MIX1 dataset is similar to the construction of UltraFeedback (Cui et al., 2023) with an additional summarization dataset. In comparison, the MIX2 consists of more reasoning preference pairs (math and code) and safety data. We also consider three different approaches to model the preference signals, including prompting in the LLM-as-a-judge manner (Zheng et al., 2023), reward modeling as the MLE of the BT reward model, and the preference model.

## B.2 Benchmark Details

- AlpacaEval-2 (Dubois et al., 2023): This benchmark focuses on single-turn conversations and consists of 805 test prompts covering various topics. The models are compared head-to-head with GPT-4-Preview (11/06) to compute the win rate. The same GPT-4 model is used as the judge. To mitigate the length bias of GPT-4, a length-control variant of the benchmark is also proposed.

- MT-Bench (Zheng et al., 2023): This benchmark is a multi-turn benchmark and includes 160 test prompts from 8 different areas. The model should first answer an initial question, and then a predefined follow-up question. The model's responses are then rated by the GPT-4 model with a scale from 1-10, and the final score is computed as the average score of two turns.

- Chat-Arena-Hard (Tianle et al., 2024): This benchmark consists of 500 test prompts from the live data in Chatbot Arena, a crowd-sourced platform for LLM evaluations. The prompts evaluate the model's ability in specificity, domain knowledge, complexity, problem-solving, creativity, technical accuracy, and real-world application. In addition to the agreement to human preference, compared with AlpacaEval-2 and MT-Bench, Chat-Arena-Hard further enjoys a clear separability among different models.

We also summarize the benchmarks we use in this project in Table 6.

## B.3 Other details

**Prompt Visualization.** We provide the visualization generated on Nomic Atlas[7] with the `nomic-embed-text-v1.5` text embedding model (Nussbaum et al., 2024)

**SFT Data List.** We collect open-sourced instruction-finetuning data for our SFT model training. The following data is included: ShareGPT (Chiang et al., 2023), Evol-Instruct (Xu et al., 2023a), SlimOrca (Lian et al., 2023b), MathInstruct (Yue et al., 2023), Magicoder-Evol-Instruct (Wei et al., 2023), GPT4-LLM (Peng et al., 2023), OrcaMath (Mitra et al., 2024), GPTeacher (Teknium1, 2023), UltraInteract (Yuan et al., 2024a).

**Offline Vanilla DPO.** We use Nectar dataset for Offline DPO. We run 1 epoch with batch size 128, learning rate 5e-7, and cosine decay scheduler.

---

[7]https://atlas.nomic.ai/

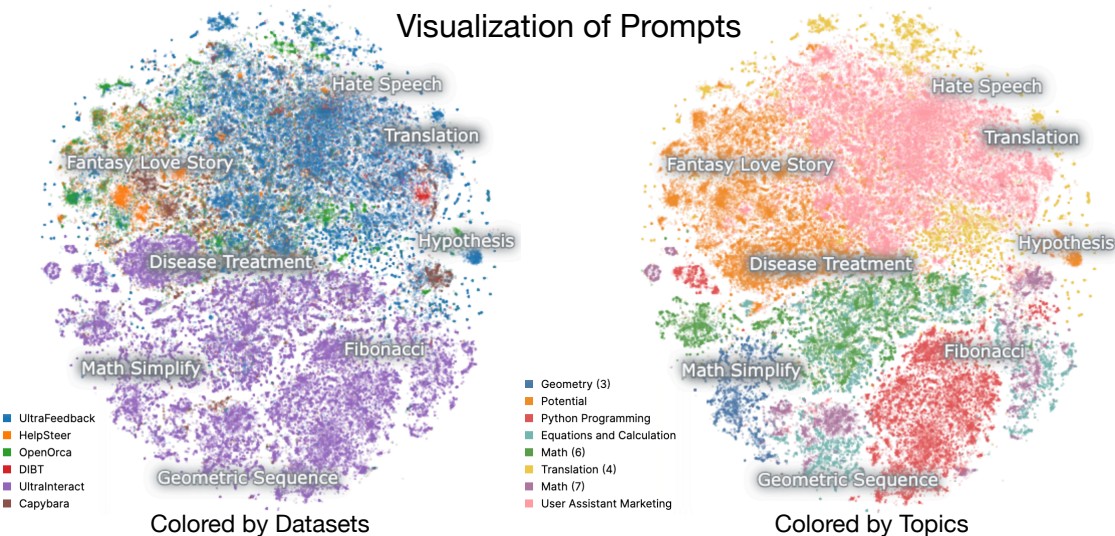

Figure 5: Visualization of our prompt collection via Nomic Atlas. The left figure is colored by the data sources of prompts, and the right figure is colored by topics, which are auto-generated by the custom topic model of Nomic Atlas.

| Parameter | Value |
|---|---|
| n_batch_size_per_device | 2 |
| n_gradient_accumulation | 8 |
| optim | adamw_torch |
| lr_scheduler_type | cosine |
| num_train_epochs | 2 |
| beta | 0.1 |

Table 7: Training parameters

**Additional Plots.** We also have some additional training plots Figure 6, 7 and have visualized our performance as Figure 9.

**Hyperparameters.** The hyperparameters are listed in Table 7.

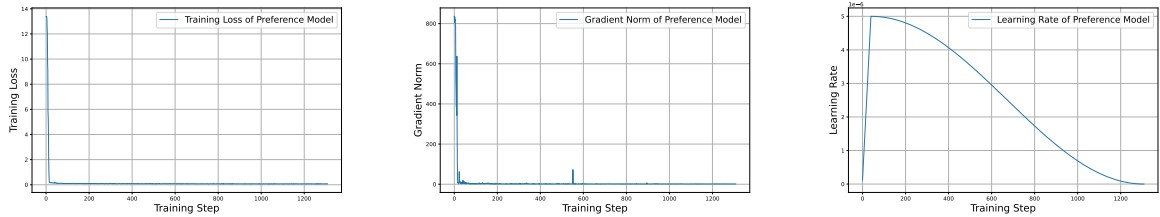

Figure 6: The training record of preference modeling. From the left to right, we present the records of training loss, gradient norm, and the learning rate, respectively.

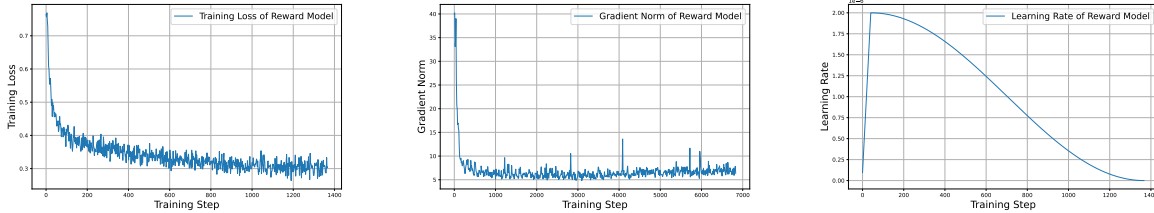

Figure 7: The training record of reward modeling. From the left to right, we present the records of training loss, gradient norm, and the learning rate, respectively.

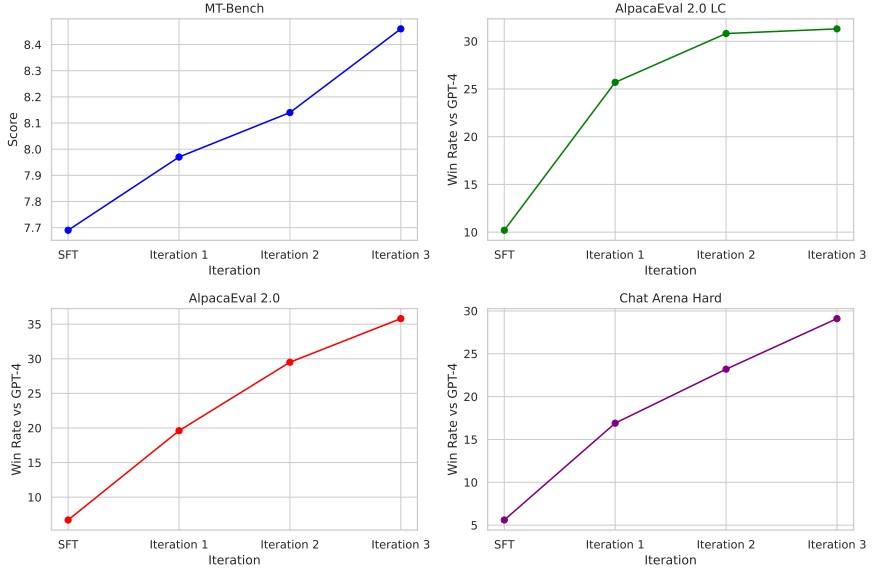

Figure 8: Model performance with respect to RLHF iterations.

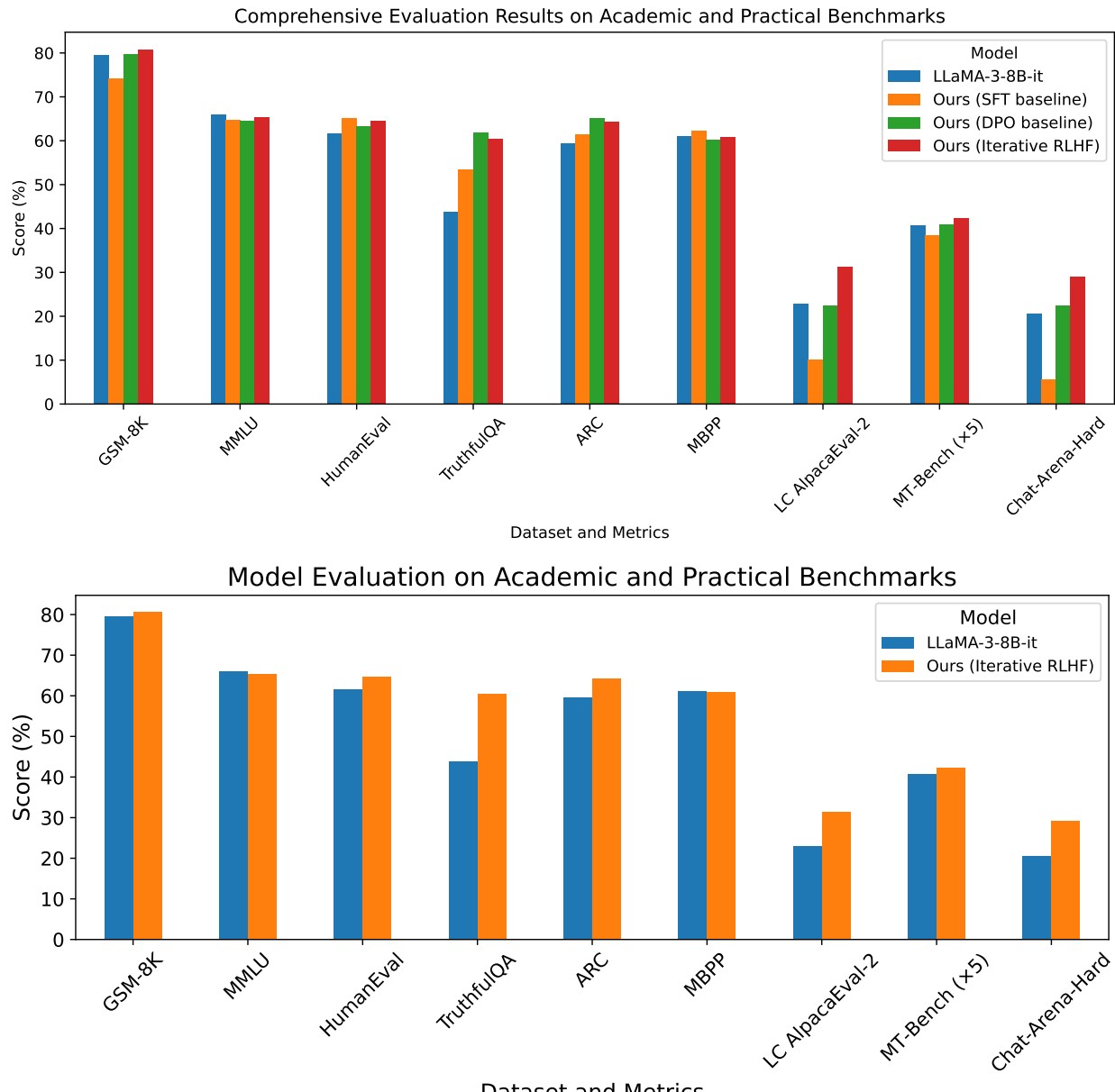

Figure 9: Evaluation of our models and LLaMA-3-8B-inst.

## C   Case Studies

To showcase the significant improvements achieved through online RLHF, we conducted a qualitative analysis of the generated responses. We observed that, after applying online RLHF, the responses became more detailed and better formatted, often utilizing bullet points, highlights, bold text, and enumeration. This enhanced clarity and structure were key factors contributing to the improved win-rate observed in our experiments. Below, we present several examples that illustrate these improvements.

---

**Prompt**

What is Gremolata?

---

**Offline RLHF Response**

Gremolata is a flavorful, aromatic mixture used as a garnish or seasoning for a variety of Italian dishes, particularly osso buco, a dish made from braised veal shanks. It is also used in other dishes like fish, pasta, or vegetables. Gremolata is typically made from a combination of finely chopped fresh parsley, garlic, and lemon zest. Sometimes, basil or mint can be added as well. The mixture is usually prepared by finely chopping or mincing the ingredients and then mixing them together. Gremolata adds a bright, fresh, and citrusy flavor to the dishes it is used in. It is often sprinkled on top of the dish just before serving, allowing the flavors to meld and enhance the overall taste.

---

**Online RLHF Response**

Gremolata is a flavorful, aromatic mixture used as a garnish or seasoning in Italian cuisine, particularly in dishes such as Osso Buco, risotto, and other meat or fish dishes. It is often served on top of the dish to add a burst of fresh flavors.
The basic ingredients of Gremolata are:
1. Fresh parsley: This is the main component, providing the fresh, herbaceous flavor and color to the mixture. Flat-leaf parsley is often preferred for its robust flavor.
2. Garlic: Finely chopped or minced raw garlic gives a sharp, pungent taste to the Gremolata.
3. Lemon zest: The zest from a lemon adds a bright, citrusy flavor and a slight tang to the mixture.
In some variations, you might also find:
4. Lemon juice: This adds a little extra acidity and freshness to the mix. 5. Fresh basil or mint: These herbs can be used instead of, or in addition to, parsley, depending on the dish and personal preference.
To prepare Gremolata, simply mix the ingredients together in a small bowl. The proportions can vary depending on personal taste, but a general guideline is 1 part lemon zest, 2 parts chopped parsley, and 12 part chopped garlic. Adjust to your liking.
Gremolata is typically added just before serving to maintain its freshness and bright flavors. It pairs well with rich and savory dishes, as it helps to brighten and lighten the overall taste.

Figure 10: Case Study – Example 1.

**Prompt**

What are the best exercises for beginners?

**Offline RLHF Response**

The best exercises for beginners are those that are simple, safe, and effective in improving overall fitness. Some of the top beginner exercises include:
1. Squats: Helps to strengthen the lower body, particularly the glutes, quadriceps, and hamstrings. 2. Push-ups: Improves upper body strength, targeting the chest, triceps, and shoulders. 3. Planks: Engages the core muscles and strengthens the lower back, arms, and glutes. 4. Lunges: Strengthens the quadriceps, hamstrings, and glutes, while improving balance and flexibility. 5. Standing Rows: A great exercise for the upper back, biceps, and shoulders. 6. Bicep curls: Targets the biceps, promoting arm strength and definition. 7. Tricep extensions: Strengthens the triceps, an important muscle for overall upper body strength. 8. Walking or jogging: A simple and effective way to improve cardiovascular fitness and burn calories. 9. Bodyweight squats and lunges: These are great for beginners as they only require bodyweight and help build strength and balance. 10. Stretching and yoga: Incorporating flexibility exercises like yoga or simple stretches can help prevent injury and promote overall mobility.
Remember to start with low weights or bodyweight exercises, gradually increasing the intensity, and always focus on proper form to avoid injury. It's also a good idea to start with a warm-up and cool-down, and consider working with a personal trainer or coach to learn the correct techniques.

**Online RLHF Response**

The best exercises for beginners are those that are simple, safe, and effective in improving overall fitness, strength, and flexibility. Here are some top recommendations for starting your fitness journey:
1. **Bodyweight Squats**: A fundamental exercise that targets the lower body, including the glutes, hamstrings, and quadriceps. Stand with feet shoulder-width apart, lower your hips as if sitting back into a chair, then push back up to the starting position. Aim for 2-3 sets of 10-15 reps.
2. **Push-ups**: A classic upper body strengthener that targets the chest, triceps, and shoulders. Start on your hands and knees, then lower your body until your chest nearly touches the ground, then push back up. Modify by starting on your knees if needed. Aim for 2-3 sets of 8-12 reps.
3. **Plank**: A core-strengthening exercise that also works the shoulders, back, and glutes. Start in a push-up position, then lower your forearms to the ground, keeping your body in a straight line from head to heels. Hold for 30 seconds to 1 minute, then rest and repeat for 2-3 sets.
4. **Lunges**: A lower body exercise that targets the glutes, hamstrings, and quadriceps. Stand with feet together, then step forward with one foot, lowering your hips until both knees are bent at a 90-degree angle. Push back to the starting position and repeat on the other side. Aim for 2-3 sets of 8-12 reps per leg.
5. **Bent-over Row**: A back and shoulder strengthener that targets the latissimus dorsi, rhomboids, and traps. Hold a dumbbell in each hand, bend forward at the hips with a straight back, then row the weights up to your sides, squeezing your shoulder blades together. Lower the weights and repeat for 2-3 sets of 8-12 reps.
6. **Walking or Jogging**: A great way to improve cardiovascular fitness and burn calories. Start with a brisk walk, gradually increasing speed and duration as you become more comfortable. Aim for at least 30 minutes of moderate-intensity exercise, such as brisk walking, 5 days a week.
7. **Yoga or Pilates**: These low-impact exercises are excellent for improving flexibility, balance, and core strength. Start with beginner classes or videos, focusing on proper form and breathing. Aim for at least 2-3 sessions per week.
8. **Calf Raises**: A lower body exercise that targets the calf muscles. Stand on the edge of a step or platform, then raise your heels as high as possible, then lower back down. Aim for 2-3 sets of 10-15 reps.
Remember to warm up before starting any exercise, and always listen to your body. If an exercise causes pain or discomfort, stop and consult a healthcare professional or fitness instructor. As you progress, you can gradually increase the intensity, duration, and variety of your workouts to continue challenging your body and achieving your fitness goals.

Figure 11: Case Study – Example 2.

**Prompt**

"What type of soil is suitable for cactus?"

**Offline RLHF Response**

Cactus plants thrive in well-draining soil that is porous and allows excess water to escape quickly. The ideal soil for cactus should have a pH level between 6.0 and 7.5, which is slightly acidic to neutral. The soil should also be low in nutrients, as cactus plants generally prefer a nutrient-poor environment.
The following soil types are suitable for cactus plants:
1. Cactus-specific soil mix: This is a commercial soil mix designed specifically for cactus plants. It typically contains a blend of sand, coconut coir, perlite, and peat moss or sphagnum moss. This mix provides excellent drainage and aeration while retaining moisture.
2. Desert soil mix: A desert soil mix can also be suitable for cactus plants. It usually consists of a blend of sand, perlite, andor vermiculite. This mix is well-draining and allows water to escape quickly.
3. Sandy soil: A sandy soil with a high percentage of sand is suitable for cactus plants. It provides excellent drainage and allows water to escape quickly.
4. Gravel and cactus mix: Layering a cactus mix on top of a layer of small rocks or gravel can also be a good option. This mimics the natural environment of cactus plants in the desert and allows water to drain away quickly.
It's essential to avoid using heavy, clay-based soils, as they can retain too much water, leading to root rot and other issues for cactus plants. Additionally, it's essential to ensure that the soil is well-aerated, as cactus plants require good air circulation around their roots.

**Online RLHF Response**

Cactus plants thrive in well-draining soil that allows excess water to pass through and does not hold moisture for long periods. The ideal soil for cactus plants is a mix that mimics their natural desert habitat. Here are some characteristics of suitable cactus soil:
1. **Coarse Texture**: The soil should have a coarse texture, allowing water to drain quickly. This is usually achieved by using materials like sand, perlite, or pumice. These materials help prevent waterlogged roots and root rot, which are common issues for cacti.
2. **Loamy Soil**: A loamy soil that is a combination of sand, silt, and clay is also suitable. It retains some moisture while still allowing for proper drainage. Adding some coarse materials like perlite or pumice to the loamy soil can enhance drainage further.
3. **Low Nutrient Content**: Cacti have shallow root systems and do not require a lot of nutrients. Therefore, a soil with low nutrient content, such as a cactus-specific potting mix, is ideal. Don't use regular garden soil, as it may have too many nutrients and can lead to excessive growth and weak stems.
4. **Acidic to Slightly Alkaline**: Cactus plants prefer slightly acidic to slightly alkaline soil, with a pH range of 6.0 to 7.5. This range is generally neutral and does not require additional adjustments.
5. **Dry and Aerated**: The soil should be dry on the surface to prevent moisture buildup, but still aerated enough to allow oxygen to reach the roots. This promotes healthy root growth and prevents issues like root rot.
To create a suitable cactus soil mix, you can use a combination of the following ingredients:
- 2 parts cactus or succulent potting mix (which usually contains peat, perlite, and coconut coir)
- 1 part coarse sand (such as sharp sand or builder's sand)
- 1 part perlite or pumice (to improve drainage)
Mix these ingredients thoroughly and use the resulting soil for your cactus plants. You can also purchase commercial cactus and succulent soil mixes from a nursery or garden center, which are usually well-suited for these types of plants.
Remember to always allow the soil to dry out completely between waterings, as cactus plants are adapted to survive in arid conditions. Overwatering is a common cause of cactus issues, so be cautious when watering your cacti.

Figure 12: Case Study – Example 3.

