# OpenReview forum: "RLHF Workflow: From Reward Modeling to Online RLHF"
_TMLR — Accepted by TMLR_

### Review · Reviewer_7Xe9 · 2024-07-02

**Summary Of Contributions:**

This paper presents an online RLHF workflow using open source data and models. Specifically, they propose to first train a proxy reward model using a mixture of public available datasets. Then, they perform iterative Direct Preference Learning to update the model. They obtain a model that has impressive performance in conversation tasks, while retain the model's ability in academic task. They also explore incorporating length penalty in reward modeling and show that it gives superior performance in length-control benchmark.

**Audience:**

Yes

**Broader Impact Concerns:**

RLHF system learns from human preference, and it is possible that existing stereotypes are reinforced during the training of such system.

The long-term consequences of deploying RLHF systems at scale are not fully understood. These systems have the potential to influence societal behaviors and decision-making processes in profound ways.

**Claims And Evidence:**

Yes

**Requested Changes:**

1. (critical) For such an online approach, it is important to see how the performance of the model changes w.r.t the number of iterations. I don't see such a plot in the current version.
2. (critical) Some case study should be included to compare offline and online RLHF approach. For example, what types of prompts are better answered by the online trained model compared to the offline one.
3. (good to have) The preference model overall has better performance according to Table 1, and it would be interesting to see some initial RL experiment results using preference model.

**Strengths And Weaknesses:**

Strength: This work gives a simple but effective online RLHF framework. It only makes use of public available datasets with no extra human labels, and yet achieves impressive performance.

Weakness: All components in this work are not novel. Experiment results align with that of existing works, and there is no new insights.

---

> ### Author Response · Authors · 2024-08-01
>
> 1. **Novelty and Insights**
>
> "Bag of tricks" papers are crucial in ML research, offering practical methods that drive progress. They facilitate rapid adoption of effective techniques, aiding both researchers and practitioners in enhancing models and maintaining competitiveness. This is particularly true for preference learning for LLMs as the big technology companies do not share 100\% of what they do and making the RLHF techniques more accessible to the community is interesting from our side.
>
> Examples include:
> - Tim Salimans et al. "Improved Techniques for Training GANs," NIPS 2016.
> - Tong He et al. "Bag of Tricks for Image Classification with CNNs," CVPR 2019.
> - Alex Nichol and Prafulla Dhariwal. "Improved Denoising Diffusion Probabilistic Models," ICML 2021.
>
> Our models and codes have supported numerous academic projects, such as:
> - Xu Zhangchen, et al. "Magpie: Alignment Data Synthesis from Scratch by Prompting Aligned LLMs with Nothing."
> - Chen, Lichang, et al. "OPTune: Efficient Online Preference Tuning."
> - Xie, Tengyang, et al. "Exploratory Preference Optimization: Harnessing Implicit Q*-Approximation for Sample-Efficient RLHF." arXiv preprint arXiv:2405.21046 (2024).
> - Zhong, Han, et al. "Dpo meets ppo: Reinforced token optimization for rlhf." arXiv preprint arXiv:2404.18922 (2024).
> - Zheng, Chujie, et al. "Weak-to-strong extrapolation expedites alignment." arXiv preprint arXiv:2404.16792 (2024).
> - Ye, Chenlu, et al. "A theoretical analysis of nash learning from human feedback under general kl-regularized preference." arXiv preprint arXiv:2402.07314 (2024).
> - Chen, Ruijun, et al. "Self-Evolution Fine-Tuning for Policy Optimization"
> - Li Bolian, et al., Cascade Reward Sampling for Efficient Decoding-Time Alignment
> - Zhang, Yuheng, et al. "Iterative Nash Policy Optimization: Aligning LLMs with General Preferences via No-Regret Learning"
> - Lin Tzu-Han, et al., "DogeRM: Equipping Reward Models with Domain Knowledge through Model Merging",
> - Yang Rui, et al., "Regularizing Hidden States Enables Learning Generalizable Reward Model for LLMs"
> - Junsoo Park, et al., "OffsetBias: Leveraging Debiased Data for Tuning Evaluators"
> - Meng Yu, et al., "SimPO: Simple Preference Optimization with a Reference-Free Reward"
>
> Additionally, TMLR guidelines state, "Nor should it form the basis for rejecting work on a method considered not ‘novel enough,’ as novelty of the studied method is not a necessary criterion for acceptance."
>
> 2. **Model Performance w.r.t Number of Iterations**
>
> We have included additional plots in the latest Appendix to address performance changes over iterations.
>
> 3. **Case Study: Offline vs. Online RLHF**
>
> Thank you for the suggestion. Please see Appendix B for a detailed case study comparing offline and online RLHF approaches.
>
> 4. **RL Experiment Results Using Preference Model**
>
> We appreciate the suggestion. Our preliminary results using the preference model during the RL stage show comparable performance to the BT-based approach:
> - Alpaca LC: 28.9
> - Chat-Arena-Hard: 31.7
> - MT-Bench: 8.40
> One main drawback of the preference based method is the computation because to perform rejection sampling for n responses, we need to conduct $O(n^2)$ pairwise comparisons. Moreover, the advantage of the pairwise preference model seems to be on the reasoning task, while we focus on making general chatbots in this work. We leave a more detailed study to future work on improving the reasoning ability of LLMs.

---

### Review · Reviewer_nVba · 2024-07-16

**Summary Of Contributions:**

This paper proposes the workflow of Online Iterative Reinforcement Learning from Human Feedback (RLHF).

**Audience:**

No

**Broader Impact Concerns:**

see above.

**Claims And Evidence:**

Yes

**Requested Changes:**

1. During reading, some confusing points arise, for example:
"The dominant RLHF framework used for the LLM alignment was first popularized in Christiano et al. (2017)."
Christiano et al. (2017) did not conduct LLM alignment?

1. The author elaborates on LLMs alignment but does not define what alignment is throughout the text?
In section 2.2, the author mentions "Length bias in reward modeling," which is an interesting point regarding RM's length bias, but the author does not define this issue or explain why their proposed method can alleviate it.

1. In this technical report, we aim to fill in this gap and provide a detailed recipe that is easy to reproduce for online iterative RLHF. What does this "recipe" specifically refer to?
discuss the theoretical insights and algorithmic principles behind online iterative RLHF; The text contains eight numbered formulas, the first six of which are common sense. What specific insights are being referred to?

Regarding **reproducibility**, the experimental parameters provided by the author are insufficient to support reproducing the algorithm in this paper.

Here are some of my points of confusion, and I am very willing to communicate further with the author.

**Strengths And Weaknesses:**

1. Understanding this paper requires extensive prior knowledge, and its writing makes it difficult to read.
> For example, the core of this paper revolves around RLHF research, especially since the author only discusses RLHF in the first paragraph: as a key technique for integrating human preference signals into machine learning methods, particularly in aligning Large Language Models (LLMs) with human values and preferences.

2. What are "human preference" and "values"? From a technical perspective, how are these two reflected? These are all prior knowledge requirements. Additionally, the author begins discussing issues with RLHF in the second paragraph without defining RLHF, which is confusing.

4. Some crucial concepts and references in the field of alignment are not discussed by the author, such as:

- The challenges faced by RLHF. e.g. Open Problems and Fundamental Limitations of Reinforcement Learning from Human Feedback
- The 3H criteria of large language models. e.g. A General Language Assistant as a Laboratory for Alignment
- The definition of alignment. e.g. AI Alignment: A Comprehensive Survey

---

> ### Author Response · Authors · 2024-08-01
>
> **Weaknesses Part**
>
> Thanks for the detailed feedback and constructive comments! The main purpose of this work is to provide a detailed, easy-to-reproduce recipe to make the online iterative RLHF pipelne more accessible to the open-source community and we agree that a through review of the development of RLHF techniques would be very helpful to position the work.
>
> The idea of learning from relative feedback can date back to the study of dueling bandit [1]. The standard RLHF was popularized by [2](Christiano et al (2017)), which served to direct the attention of the deep RL community to the preference-based feedback. Then, the techniques was further introduced to fine-tune LLMs in some early works [3,4].
>
> The most popular framework for the RLHF of modern LLMs is developed and outlied in [5] (GPT) and [6] (Claude), where they present a two-stage method: reward modeling based on MLE of BT model and policy optimization based PPO. In particular, [6] proposes to make LLMs helpful, harmless, and honest.
>
> After the launch of ChatGPT, RLHF has received considerable research interests. However, the instability of the PPO training and the challenging of infra and GPU memory resource prevent the opensource community from reproducing the success of the closed-source RLHF models. In view of this, the open-source community has proposed alteranative approach for the LLM alignment. In particular, a line of work has designed algorithms in a direct preferenec learning style, including [7-10]. These algorithms directly optimize the policy by optimizing a contrastive loss on the preference dataset without constructing a reward model first (hence the name direct preference learning). Among them, DPO is the most popular one and has been widely adopted in both industry and academia nowadays.
>
> The open-source project Zephyr[11] serves as a milestone to popularize the DPO, where the authors presented a recipe to do offline and offpolicy DPO through distillation from a teacher model (ChatGPT). Using that recipe, many open-source models are fine-tuned by offline DPO and their qualities are largely improved compared to their SFT counterparts.
>
> After this, [12,13] propose online iterative DPO, where an external reward model/preference is used to label the responses generated by the model itself (or its variant). The authors present initial evidences to support the effectiness of the proposed approach. However, the existing open-source reward models were still largely sub-optimal and a recipe to make the state-of-the-art open-source model by online iterative DPO was also missing.  Indeed, the LLaMA2-13B-based RM was the SOTA open-source model at the beginning of our project, which was largely behind the models we obtain in this work. To fill in this gap, we present this work. Since the first release of this work, our models and codes have supported numerous academic projects, such as:
> - Xu, et al. "Magpie: Alignment Data Synthesis from Scratch by Prompting Aligned LLMs with Nothing."
> - Chen, et al. "OPTune: Efficient Online Preference Tuning."
> - Xie, et al. "Exploratory Preference Optimization: Harnessing Implicit Q*-Approximation for Sample-Efficient RLHF."
> - Zhong, et al. "Dpo meets ppo: Reinforced token optimization for rlhf."
> - Zheng, et al. "Weak-to-strong extrapolation expedites alignment."
> - Ye, et al. "A theoretical analysis of nash learning from human feedback under general kl-regularized preference."
> - Chen, et al. "Self-Evolution Fine-Tuning for Policy Optimization"
> - Li, et al., Cascade Reward Sampling for Efficient Decoding-Time Alignment
> - Zhang, et al. "Iterative Nash Policy Optimization: Aligning LLMs with General Preferences via No-Regret Learning"
> - Lin, et al., "DogeRM: Equipping Reward Models with Domain Knowledge through Model Merging",
> - Yang, et al., "Regularizing Hidden States Enables Learning Generalizable Reward Model for LLMs"
> - Junsoo, et al., "OffsetBias: Leveraging Debiased Data for Tuning Evaluators"
> - Meng, et al., "SimPO: Simple Preference Optimization with a Reference-Free Reward"
>
> [1] The K-armed Dueling Bandits Problem
>
> [2] Deep reinforcement learning from human preferences.
>
> [3] Fine-tuning language models from human preferences.
>
> [4] Learning to summarize with human feedback.
>
> [5] Training language models to follow instructions with human feedback.
>
> [6] Training a helpful and harmless assistant with reinforcement learning from human feedback.
>
> [7] Slic-hf: Sequence likelihood calibration with human feedback
>
> [8] Direct preference optimization: Your language model is secretly a reward model
>
> [9] A general theoretical paradigm to understand learning from human preferences
>
> [10] Kto: Model alignment as prospect theoretic optimization.
>
> [11] Zephyr: Direct Distillation of LM Alignment
>
> [12] Iterative preference learning from human feedback: Bridging theory and practice for rlhf under kl-constraint
>
> [13] Some things are more cringe than others: Preference optimization with the pairwise cringe loss

---

> > ### Author Response · Authors · 2024-08-01
> >
> > **Requested Changes:**
> > > 1. During reading, some confusing points arise, for example: "The dominant RLHF framework used for the LLM alignment was first popularized in Christiano et al. (2017)." Christiano et al. (2017) did not conduct LLM alignment?
> >
> > Thanks for pointing this out! The Christiano et al. (2017) did not conduct LLM alignment but the current RLHF framework used for LLM alignment was developed in this project: trajectories and preference signals colelction; reward modeling and policy optimization by deep RL method.
> >
> > > 2. The author elaborates on LLMs alignment but does not define what alignment is throughout the text? In section 2.2, the author mentions "Length bias in reward modeling," which is an interesting point regarding RM's length bias, but the author does not define this issue or explain why their proposed method can alleviate it.
> >
> > The alignment problem is eventually abstracted as a KL-regularized reward optimization problem in equation (2) in the original version. We have add a more comprehensive review of the development of the RLHF techniqeus in the literature as we described in the response of weakness part.
> >
> > We would also like to clarify that we do desribe the notion of length bias in reward modeling though we do not define it in a mathemartically rigorous way. See the paragraph starting with Length bias in reward modeling in page 6. The reason why our proposed method can alleviate this bias is becasue we explicitly subtract $\lambda |y|$ in the reward signal where |y| is the character length of the response and $\lambda$ is the penalty coefficient.
> >
> >
> > > In this technical report, we aim to fill in this gap and provide a detailed recipe that is easy to reproduce for online iterative RLHF. What does this "recipe" specifically refer to? discuss the theoretical insights and algorithmic principles behind online iterative RLHF; The text contains eight numbered formulas, the first six of which are common sense. What specific insights are being referred to?
> >
> > The main purpose of this work is to provide a detailed guidance to make the online iterative RLHF pipeline more accessible to the open-source community so that others can easily reproduce. The main idea is to do the online iterative RLHF with strategical explorations, instead of the offline and off-policy one (learning from the responses collected by Chat-GPT) presented in the original Zephyr project. However, the original theoretical algorithms are not directly applicable so we discuss some reasonable approximation of the exploration strategy, as we detail in section 3.2 and 3.3. Then, we present a complete pipeline, starting from the pre-trained model, to make the open-source general chatbots in section 4.
> >
> >
> > > Regarding reproducibility, the experimental parameters provided by the author are insufficient to support reproducing the algorithm in this paper.
> >
> > Thanks for this point. We attach our code with the revision. We notice that many follow-up works (For example, Iterative Nash Policy Optimization: Aligning LLMs with General Preferences via No-Regret Learning) have used our models or reproduce the reuslts in this paper.

---

### Review · Reviewer_YMeA · 2024-07-21

**Summary Of Contributions:**

The paper aims to provide a recipe/workflow for online iterative RLHF methods and reward modeling (which aims to replace ground truth reward function).

The paper first provides two ways of reward model modeling: one through the BT model construction (a more classical approach), and the second one trhough preference model construction (a more recent approach). The paper compares the two constructions over the Reward-Bench benchmark.

The paper then introduce the online iterative RLHF recipe. They start from the theoretical framework from Xiong el al., 2023, and then provide practical algorithms to replace the part with computational infeasibility. Specially, the paper provides how to combine the offline data and the way to collect and label online data for the interative RLHF procedure. Finally, the experiment result across a comprehensive benchmarks demonstrate the effectiveness of the iterative methods.

**Audience:**

Yes

**Claims And Evidence:**

Yes

**Requested Changes:**

1. adding the discussion with [1,2,3] (however, I am not sure if the submission is earlier or [1], please disregard if the submission is earlier).
2. explaining the connection between the current data collection method and exploration.
3. fix the minor issues.
4. adding hyperparameter table and algorithm box for the practical version.

**Strengths And Weaknesses:**

## Strength

1. The iterative online methods is a very natural idea and the paper provides a relatively principled framework for this idea.
2. The reward modeling part is very significant for any online RLHF method and the paper does a decent effort investigating the rewad modeling aspect, e.g., different reward model contruction and length control.
3. The experiment result is comprehensive and demonstrates the benefit of the iterative online method.

## Weakness

1. The preference model construction of the reward model seems strange. I might have missed something, but if we follow this construction, doesn't it mean that it is possible $\hat P(a^1 \prec a^2 \mid x,a^1,a^2) + \hat P(a^2 \prec a^2 \mid x,a^1,a^2) < 1$? Does it not matter in practice or with fair amount of training we can ensure the support of the token after "label =" is only on "A" and "B" with good generalization?
2. As the paper also mentions, the exploration scheme in line 4 of Alg. 1 is generally computationally intractable. I also wonder why we need such a relatively complicated exploration strategy. As [1] proves, using forward KL as exploration bonus seems to suffice even in general function approximation setting?
3. It is hard to tell if the current suggested exploration strategies are actually performing exploration -- there seems no active exploration in these methods. For example, labeling from samples of different checkpoint does not lead to exploration, it is more like reweighting/filtering on the current distributions?
4. It will be beneficial to contextualize the results and contributions by comparing and contrasting with hybrid learning in general RL, e.g., [2,3] by comparing the settings and desiderata.

## Minor
1. Honestly, I did not get why the current figure 3 imply that the reward model is biased towards longer reponses. Maybe more explaination could be helpful.
2. what do mix1 and mix2 mean in table 1?
3. In the main text, it says fig.3 is generated with 8 reponses per prompt, but the caption says 16.

[1] Exploratory Preference Optimization: Harnessing Implicit Q*-Approximation for Sample-Efficient RLHF

[2] Policy finetuning: Bridging sample-efficient offline and online reinforcement learning

[3] Hybrid rl: Using both offline and online data can make rl efficient

---

> ### Author Response · Authors · 2024-08-01
>
> > 1. The preference model construction of the reward model seems strange. I might have missed something, but if we follow this construction, doesn't it mean that it is possible? Does it not matter in practice or with fair amount of training we can ensure the support of the token after "label =" is only on "A" and "B" with good generalization?
>
> Thanks for this point! We should have described the pipeline of preference model with more details and we would adjust the report accordingly. We actually comduct a re-normalization on the probabilities of decoding A or B when we use the model. But we also notice that for almost all the cases, the probabilities are concentrating on the two tokens A and B.
>
> > 2. As the paper also mentions, the exploration scheme in line 4 of Alg. 1 is generally computationally intractable. I also wonder why we need such a relatively complicated exploration strategy. As [1] proves, using forward KL as exploration bonus seems to suffice even in general function approximation setting?
> 3. It is hard to tell if the current suggested exploration strategies are actually performing exploration -- there seems no active exploration in these methods. For example, labeling from samples of different checkpoint does not lead to exploration, it is more like reweighting/filtering on the current distributions?
>
>
> Actually, [1] is released after the submission of this work, and we are aware of two more independent works considering similar idea. However, we would be happy to make a discussion on the design of exploration to make the ideas of our work clear. The idea of using a ``feel-good'' loss function with optimistic bias was first proposed in [i] and its extended version [ii]. From a theoretical perspective, this is an alternative exploration strategy that **does not offer any advantage in terms of theoretical convergence rate in its original application to RL with general function approximation.** Meanwhile, while the ''feel-good'' objective seems to be easier to approximate in practice, it is non-convex in general (due to the presense of the biased term). Therefore, while it is simple, eseentailly it does not simplify the problem (see the discussions of [i] around page 9). Moreover, we remark that extending the original result established in the linear function approximation case to the general function approximation is relatively trivial given all the techniques established in the past few years of RL theory study. See the follow-up work of the authors of the original paper [iii].
>
> To summarize, [i] (and some other concurrent works) presents another alternative way to do exploration, thus consistent with the main intuition of this work: we need strategical exploration (beyond pure on-policy sampling) to improve the model performance. We will cite these works in the discussion of potential approaches to do exploration. However, we would also like to mention that in the context of general neural network models, how to do exploration is still an open problem and the theoretical insights can only be applied to guide the empirical study *in principle*.
>
> In view of this, the principle of exploration (equation 8) is interpreted as to increase the data diversity while maintaining a moderate KL divergence between the current MLE policy (obtained by DPO). We consider sampling from different checkpoints, and rejection sampling the current some known aprpoxiamtions that work well in practice. The alternative approach in [i] is also a possibility. But from the empirical side, they are constrained to play against a fixed sampling policy, and their experimental results also do not really outperform ours. We hope that there can be more works to study the uncertainty estimation and exploration in the context of LLMs in the future.

---

> > ### Author Response · Authors · 2024-08-01
> >
> > > 4. It will be beneficial to contextualize the results and contributions by comparing and contrasting with hybrid learning in general RL, e.g., [2,3] by comparing the settings and desiderata.
> >
> > We are aware of these two works that focus more on the theoretical techniques. The focus of this wrok is to present a recipe for online iterative RLHF and to make these techniques more accessible to the open-source community. In particular, we follow the work [iii] to formulate the problem as a hybrid learning for the sake of generality but we do not focus on the impact of offline dataset. However, we would be happy to make some discussions on them.
> >
> > In comparison, the work [3] is more related to the original paper [iii] in terms of analysis technique though the authos of [iii] and the authors of [3] consider different tasks. The core idea of [3] is that the suboptimality can be decomposed into two parts, one related to $\pi^*$, and another one related to $\hat{\pi}_t$ (see equation 3.2 of [4]). In the previous RL theory study, they are handle seprately: for online setting, we use optimism to handle $\pi^*$ term and use exploration condition (eluder dimension, elliptical potential lemma for linear function approximation or bilinear class); for offline setting, we use pessimism to handle $\hat{\pi}$ and use data coverage of the dataset to handle $\pi^*$. However, the optimism and pessimism are exactly the reason why the theoretical algorithms are not practical. Therefore, they propose that if we have a good offline dataset with a good coverge on $\pi^*$, and in the meantime, the problem satisfies the exploration condition. Then, no optimism or pessimism is required.
> >
> > We notice that this is a very clever observation made in [3] but is not specialized for LLMs. In particular, as we discuss at the end of section 1.1, in the context of LLMs, due to the large exploration shift, it is unlikely that we can have an offline dataset with a good coverage. Though the situation can be different because we are concerning the KL-regularized target so we are searching the optimal policy within a KL ball of the original policy, the study would be more complicated and is beyond the scope of this work. Therefore, we do not dive into the details of the impact of the offline dataset in this work.
> >
> >
> >
> > **Minor**
> > > Honestly, I did not get why the current figure 3 imply that the reward model is biased towards longer reponses. Maybe more explaination could be helpful.
> >
> > Thanks for the suggestion! We compute the Pearson correlation coefficeint between the reward and response length for the two reward models and plot the heatmap in the Figure 3. We can see that the samples are more concentrated on the right-hand side of the map, which means that for most of the samples, the rewards are positively correlated with the response length. Therefore, the reward model is biased towards longer response. We will add some explainations accordingly!
> >
> > > what do mix1 and mix2 mean in table 1?
> >
> > Thanks for this point. We should explain the dataset combination mix1 and mix2 in the main body of the work. Mix1 consists of HH-RLHF + SHP + UltraFeedback + Summarization, which is the popular choice in the previous work. Mix2 consists of all datasets in Table 5, which is our final dataset combination.
> >
> > > In the main text, it says fig.3 is generated with 8 reponses per prompt, but the caption says 16.
> >
> > Thanks for pointing this out! It should be 16 and we will fix this typo accordingly.
> >
> > [1] Exploratory Preference Optimization: Harnessing Implicit Q*-Approximation for Sample-Efficient RLHF
> >
> > [2] Policy finetuning: Bridging sample-efficient offline and online reinforcement learning
> >
> > [3] Hybrid rl: Using both offline and online data can make rl efficient
> >
> > [4] Is Pessimism Provably Efficient for Offline RL?
> >
> >
> > **Requested Changes:**
> > > 1. adding the discussion with [1,2,3] (however, I am not sure if the submission is earlier or [1], please disregard if the submission is earlier).
> >
> > See the discussion on the weakness 2 and 3.
> >
> > > 2. explaining the connection between the current data collection method and exploration.
> >
> > See the discussion on the weakness 2 and 3.
> >
> > > 3.fix the minor issues.
> >
> > Yes, we have fixed them accordingly in the revision.
> >
> > > 4. adding hyperparameter table and algorithm box for the practical version.
> >
> > Thanks for the suggestion! We have added them in the revision.
> >
> >
> > [i] A Sufficient Condition of Sample-Efficient Reinforcement Learning with General Function Approximation
> > [ii] Maximize to explore: One objective function fusing estimation, planning, and exploration
> > [iii] Iterative preference learning from human feedback: Bridging theory and practice for rlhf under kl-constraint
> > [iv] Online Iterative Reinforcement Learning from Human Feedback with General Preference Model

---

### Decision · Action_Editor_LZ61 · 2024-09-04

**Recommendation:** Accept with minor revision

**Comment:**

The paper gives a detailed workflow/recipe for online iterative RLHF methods, inspired by a theoretical framework. It also includes two approaches to reward modeling, one based on the Bradley-Terry model, the other based on preference model. The workflow comes with comprehensive experiments, and demonstrates benefits of iterative online methods. The topic is important and relevant to a fast-growing community. While some of the key findings may not be novel, the detailed recipe is a useful contribution to the community. The initial submission raised questions about (minor) technical issues and related work. The revised version addressed these concerns satisfactorily. That said, I encourage the authors to polish the writing thoroughly (cf, comments by reviewer nVba). A few more detailed comments:

* The paper makes a useful distinction of preference learning and deep RL, as two approaches to RLHF, and focuses on the former. Is it more accurate to change the framing (eg, title) from RLHF to direct preferencd learning?
* The paper refers to Yue et al (2012) as the origin of using relative feedback. In fact, the idea can be traced back even further to the learning-to-rank literature: https://doi.org/10.1145/1229179.1229181
* In eqn (8), where is \Gamm_t^m defined?
* Page 9: ‘feed-good’ → ‘feel-good’?

**Audience:**

The paper is interesting to the fast-growing audience of LLM alignment and RLHF.

**Claims And Evidence:**

The paper give sufficient empirical evidence to support the claims.

---

> ### Author Response · Authors · 2024-09-20
>
> Dear AE,
>
> Thank you for your valuable comments and suggestions. We have revised the manuscript accordingly. Regarding the title, we have added a subtitle to provide clearer framing, as the original title is widely used.
>
> Best regards,
>
> Authors